# Sporadic sodium layer: A possible tracer for the conjunction between the upper and lower atmospheres

*Shican Qiu[1,2,3], Ning Wang[1,4,5], Willie Soon[6], Gaopeng Lu[2], Mingjiao Jia[2,3], Xianghui Xue[2,3], Tao Li[2,3],*

*Xingjin Wang[2,3], Xiankang Dou[2,3]*

[1]Department of Geophysics, College of the Geology Engineering and Geomatics, Chang'an University, Xi'an, 710054, China

[2]Key Laboratory of Geospace Environment, Chinese Academy of Sciences, University of Science & Technology of China, Hefei, Anhui, 230026, China

[3]Mengcheng National Geophysical Observatory, School of Earth and Space Sciences, University of Science and Technology of China, Hefei, Anhui, 230026, China

[4]Gravity & Magnetic Institute of Chang'an University, Xi'an, 710054, China

[5]Key Laboratory of Western China's Mineral Resources and Geological Engineering, China Ministry of Education, Xi'an, 710054, China

[6]Center for Environmental Research and Earth Sciences (CERES), Salem, Massachusetts, 01970, USA and Institute of Earth Physics and Space Science (ELKH EPSS), H-9400, Sopron, Hungary

*Correspondence to*: Shican Qiu (*scq@ustc.edu.cn*) *and Xiankang Dou (dou@ustc.edu.cn)*

**Abstract.** In this research, we reveal the inter-connection between lightning strokes, reversal of the electric field, ionospheric disturbances, and a sodium layer ($Na_S$), based on the joint observations by two lidars, an ionosonde, an atmospheric electric mill, a fluxgate magnetometer, and the World Wide Lightning Location Network (WWLLN). Our results suggest that lightning strokes would probably have an influence on the ionosphere and thus affecting the occurrence of $Na_S$, with the overturning of electric field playing an important role. Statistical results reveal that the sporadic E layers ($E_S$) could hardly be formed or maintained when the atmospheric electric field turns upward. A conjunction between the lower and upper atmospheres could be established by these inter-connected phenomena, and the key processes could be suggested as follows: lightning strokes→overturning of electric field→depletion of $E_S$/generation of $Na_S$.

**Keywords**: sporadic sodium layers, sporadic E layers, atmospheric circuit, lightning stroke, electric field

## 1 Introduction

The upper mesosphere-lower thermosphere (MLT) is the interface region for momentum and energy exchanges between the Earth's low atmosphere and outer space. However, on account of the limitations of detection methods, this region remains the least known part of our planet's atmosphere (Wang, 2010). Fortunately, the metal layers (especially the sodium layer), which located between about 80 ~ 110 km, could possibly act as a window to detect the MLT parameters by means of fluorescence resonance lidars (Gardner et al., 1986; Gong et al., 2002; Gong et al., 1997). With an active chemical property and high abundance of sodium atoms, the sodium layer has been widely observed and studied all over the world (Marsh et al., 2013; Collins et al., 2002; Plane, 2003; Plane et al., 1999). The sporadic sodium layer (SSL or $Na_S$), with the neutral sodium density that could double within several minutes, is the most fantastic phenomenon observed from the sodium layer. Since first reported in 1978 (Clemesha et al., 1978), many mechanisms, involving meteor injection (Clemesha et al., 1980), dust reservoir (von Zahn et al., 1987), recombination of ions and electrons in sporadic E layer ($E_S$) (Cox and Plane, 1998), and high temperature theory (Zhou et al., 1993), have all been proposed. Because the $Na_S$ is suggested to have a connection to so many atmospheric parameters, the metric or phenomenon could be appropriate in acting as a tracer for studying inter-connection between the middle and upper atmospheres. Up to now, a large number of observations report a diversity of the $Na_S$ features, but the exact mechanism for $Na_S$ is probably still uncertain (Collins et al., 2002; Cox et al., 1993; Daire et al., 2002; Gardner et al., 1995; Qiu et al., 2015; Zhou and Mathews, 1995; Zhou et al., 1993).

Among all the proposed mechanisms, the $E_S$ theory is supported by abundant observations and results from numerical simulations (Cox and Plane, 1998; Daire et al., 2002; Dou et al., 2009; Dou et al., 2010; Gardner et al., 1993; Gong et al., 2002; Kane et al., 2001; Kane et al., 1993; Kane et al., 1991; Kirkwood and Nilsson, 2000; Kwon et al., 1988; Mathews et al., 1993; Miyagawa et al., 1999; Nagasawa and Abo, 1995; Nesse et al., 2008; Shibata et al., 2006; Williams et al., 2006). The key process of $E_S$ theory is the recombination of ions and electrons in the $E_S$ layer while descending to lower altitudes (Cox and Plane, 1998; Daire et al., 2002). The $E_S$ layer is mainly influenced by the vertical wind shear (Abdu et al., 2003; Clemesha et al., 1998; Haldoupis et al., 2004; Mathews, 1998; Šauli and Bourdillon, 2008; Wakabayashi and Ono, 2005), the geomagnetic field (Resende et al., 2013; Resende and Denardini, 2012; Zhang et al., 2015; Denardini et al., 2016), and the electric field (Abdu et al., 2003; Damtie et al., 2003; Haldoupis et al., 2004; Kirkwood and Nilsson, 2000; Kirkwood and von Zahn, 1991; Macdougall and Jayachandran, 2005; Matuura et al., 2013; Nygren et al., 2006; Parkinson et al., 1998; Takahashi et al., 2015; Voiculescu et al., 2006; Wakabayashi and Ono, 2005; Wan et al., 2001; Wilkinson et al., 1993). In the Northern Hemisphere, the $E_S$ layer would descend to a lower altitude during southward electric field (Abdu et al., 2003; Damtie et al., 2003; Haldoupis et al., 2004; Kirkwood and Nilsson, 2000; Kirkwood and von Zahn, 1991; Macdougall and Jayachandran, 2005; Nygrén et al., 2006; Parkinson et al., 1998; Takahashi et al., 2015; Voiculescu et al., 2006;

Wakabayashi and Ono, 2005; Wan et al., 2001; Wilkinson et al., 1993), and observations in the polar cap suggest the electric field reversal have an influence on the probability of $E_S$ occurrences (Macdougall and Jayachandran, 2005).

On the other hand, the atmospheric electric circuit is a closed loop (Driscoll et al., 1992; Jánský and Pasko, 2014; Lv et al., 2004; Roble and Hays, 1979; Rycroft and Harrison, 2012; Rycroft et al., 2000; Suparta and Fraser, 2012; Tinsley, 2000), like a capacitor with a positive plate (e.g., the ionosphere) and a negative plate (e.g., the ground), and dielectric medium between them (e.g., the neutral atmosphere). Then the global atmospheric electric circuit formed in the capacitor, with the lightning phenomena generating an upward current (with the atmospheric electric field intensity $E<0$) and returning a downward current ($E>0$) under fair weather condition. Nowadays increasing and emerging evidences are pointing to the close link between the upper atmosphere (e.g., the positive plate) and lower atmosphere (e.g., the negative plate) (Harrison et al., 2010; Rycroft, 2006). For example, thunderstorm occurring in the lower atmosphere is suggested to have a direct impact on the $E_S$ layer based on recent observational results (Bortnik et al., 2006; Christos, 2018; Cummer et al., 2009; Curtius et al., 2006; Davis and Johnson, 2005; Davis and Lo, 2008; England et al., 2006; Fukunishi et al., 1996; Girish and Eapen, 2008; Haldoupis et al., 2012; Immel et al., 2013; Kumar et al., 2009; Kuo and Lee, 2015; Lay et al., 2015; Mangla et al., 2016; Maruyama, 2006; Pasko et al., 2002; Rodger et al., 2001; Rycroft, 2006; Sátori et al., 2013; Sentman and Wescott, 1995; Shao et al., 2013; Sharma et al., 2004; Su et al., 2003; Surkov et al., 2006; Yu et al., 2015) or even the sodium layer (Yu et al., 2017). The possible carriers or phenomena connecting the thunderstorm to the upper atmosphere are suggested to be atmospheric tides (England et al., 2006; Haldoupis et al., 2004; Immel et al., 2013), planetary waves (Lv et al., 2004), gravity waves (Davis and Johnson, 2005; Kumar et al., 2009; Lay et al., 2015; Shao et al., 2013), transient luminous event (TLEs) (Cummer et al., 2009; Fukunishi et al., 1996; Haldoupis et al., 2012; Pasko, 2008; Pasko et al., 2002; Sentman and Wescott, 1995; Sharma et al., 2004; Su et al., 2003), the solar activity (Zhang et al., 2020), and also the electric fields (Bortnik et al., 2006; Davis and Johnson, 2005; Davis and Lo, 2008; Immel et al., 2013; Kuo and Lee, 2015; Maruyama, 2006; Rycroft, 2006; Sátori et al., 2013; Shao et al., 2013).

In this research, we apply five joint observations for our case studies and statistical works: (1) Two lidars at Hefei (31.8ºN, 117.2ºE), providing observations of sodium density, mesopause temperature and zonal wind; (2) An ionosonde in Wuhan (30.5°N,114.6°E), detecting the $E_S$ and ionospheric echoes in different modes; (3) An atmospheric electric mill (30.5ºN, 114.5ºE), giving simultaneous electric field variations; (4) A fluxgate magnetometer (30.5ºN, 114.5ºE), probing the H, D, and Z magnetic field components; and (5) The World Wide Lightning Location Network (WWLLN), observing the location and power of a lightning stroke. The purpose of this study is to examine the possibility of $Na_S$ acting as a practical robust tracer for the conjunction between the upper and lower atmospheres. Our results suggest that lightning strokes may have an influence on the lower ionosphere leading to the occurrence of $Na_S$, with the atmospheric electric field probably playing an important role.

## 2 Observations and results

### 2.1 An Na$_S$ event during the overturning of electric field

A sporadic sodium layer is detected on the night of June 3$^{rd}$, 2013, by the Temperature/Wind (T/W) lidar (Li et al., 2012). The peak density observed by the west beam of the narrowband lidar is 8650 cm$^{-3}$. This Na$_S$ event occurs much higher above the centroid height of sodium layer (normally at about 92 km) (Qiu et al., 2016). Fig. 1a shows the sodium density begins to increase at about 13:20 UT, while the largest intensity of sodium enhancement occurs from about 14:20 UT, with a peak density located around 97.65 km at 14:37 UT. The simultaneous temperature observation by the T/W lidar reveals this Na$_S$ occurs in a cold region (Fig. 1b), so the high temperature mechanism appears to be inapplicable for this event.

On the other hand, the zonal wind exhibits a suitable wind shear for creating those sporadic E layers, with a westward wind above and an eastward wind below (Fig. 1c). The E$_S$ layer is predicted to form around the border of the wind shear. Observations by the ionosonde at Wuhan indeed show active sporadic E layers on that day (Fig. 2a and b). The E$_S$ series keep travelling/propagating downward starting around 6:30 UT, and then disappear at last while the Na$_S$ occurs coincidently on about 13:20 UT. Thus this Na$_S$ is better explained by the E$_S$ mechanism, in accord with our previous study which shows that a Na$_S$ higher than 96 km tends to be controlled by the E$_S$ mechanism (Qiu et al., 2016). Although the content of sodium ions in E$_S$ layers seemed to have insufficient concentration (von Zahn et al., 1989), it has also been proposed that the ions could be concentrated by the wind shear effectively (Clemesha et al., 1999; Cox and Plane, 1998; Nesse et al., 2008). On the other hand, laboratory results show that the ligand complexes of Na$^+$·X would form and thus speed up the recombination of ions in the mesopause condition (Collins et al., 2002; Cox and Plane, 1998; Daire et al., 2002). The calculated reaction rate suggests the formation of cluster ions is enhanced at lower temperatures, in accordance with the cold region observed in Fig. 1c where the sporadic sodium layer occurs.

More details about the atmospheric parameters are shown in Fig. 3. The time series of sodium density on the peak height display a sharp enhancement from 14:20 UT (marked by the vertical red dashed line in Fig. 3a). The atmospheric electric field detected by the mill exhibits an overturning at around 14:20 UT, alternating from downward direction to upward (Fig. 3b). It can be clearly observed that the enhancement of sodium density occurs coincidently with the overturning of electric field, as highlighted by the vertical red dashed line in Fig. 3. A nearby fluxgate magnetometer provides the horizontal magnetic field $H$ (nT) (Fig. 3c), showing disturbances at 14:15 UT. The total magnetic intensity $B$ could be deduced by the $H$, $Z$, and $D$ components ($B = \sqrt{H^2 + D^2 + Z^2}$) from fluxgate magnetometer observations (the calculated values are plotted in Fig. 3d).

It is worth noting that the overturning of atmospheric electric field discussed here is theoretically rough, since the electric field at the lower ionosphere will be modulated as well (e.g., with a value of several mV/m [Seyler et al., 2004)]). Nevertheless, model simulations from the electrodynamics show that the upward electric field in upper atmosphere is proportional to the source current in the troposphere (Driscoll et al., 1992), and that the upward current would continue

transmitting to the heights of 100~130 km of the dynamo region where $E_S$ occurs most frequently (Rycroft et al., 2012). The model, based on rocket observations, shows that the atmospheric electric field has a similar scale and the same polarity from the ground to the altitude of ionosphere (Abdu et al., 2003). Thus, the electric field detected by a ground-based mill could reasonably be a reflection of the actual situation in the lower ionosphere, at least for the trends and tendencies of variations.

## 2.2 Statistical results of the $E_S$ and overturning of electric field

A statistical correlation between the overturning of electric field and $E_S$ variations is summarized in Table 1. The $foE_S$ values refers to the Critical Frequency, at which the reflection starts while the radio frequency equals the plasma frequency ($foE_S = \frac{\omega_{pe}}{2\pi} = \left(\frac{n_e e^2}{4\pi^2 m_e \varepsilon_0}\right)^{\frac{1}{2}} \approx 9\sqrt{10^{-6} n_e}$ , with $foE_S$ in MHz and $n_e$ in cm$^{-3}$) (Bittencourt, 2004). From 2012 to 2014, in the summer season (from May to August, when the $E_S$ layers occur frequently), there are 242 days with effective observations of $E_S$ and electric field with the overturning feature. Among all the cases, for about 155/242 days the $foE_S$ get interrupted during the overturning of electric field, while the $foE_S$ decreases on 39/242 days. In comparison, $foE_S$ appears only for 6/242 days, and increases for 26/242 days. Another 28/242 days show no distinguishing features of the $E_S$ during an overturning of the electric field. These ratios are overlapped, since sometimes there are more than one overturning on a single day. Thus in general, these results suggest that $E_S$ could hardly be formed for upward electric field situations and that indeed the overturning of electric field causes a depletion of $E_S$. This statistical result may also give an implication that the electric field variations detected by ground based mill have a feasible link with the lower ionosphere. It is worth mentioning that there is indeed no one-to-one correspondence between the electric field overturning and $E_S$ depletion, since the $E_S$ is also influenced by other key parameters and phenomena including wind shears, tides and gravity waves. Sometimes the overturnings recover quickly, without sufficient time for producing an obvious effect on $E_S$. Therefore a deeper analysis will be needed in future statistical studies.

## 3 Discussions

### 3.1 Possible influences by the atmospheric electric circuit

The atmospheric electric circuit is formed by the ionosphere and ground surface with the dielectric medium (e.g., the neutral atmosphere) sandwiched between them (Driscoll et al., 1992; Harrison, 2020; Jánský and Pasko, 2014; Lv et al., 2004; Roble and Hays, 1979; Rycroft and Harrison, 2012; Rycroft et al., 2000; Rycroft et al., 2012; Rycroft et al., 2007; Suparta and Fraser, 2012; Tinsley, 2000). The lightning phenomena and thunderstorms, acting as the electric generator for the circuit, drive an upward current to the ionosphere. In fair weather regime, the electric field directs downward to the earth surface (E > 0), making a closed global electric circuit (see Fig. A1 in Appendix). The electric field could vary through two

distinct ways as follows: The first one is the changing magnetic field explained by the Faraday's law (e.g. , $\nabla \times \vec{E} = -\frac{\partial \vec{B}}{\partial t}$). However, observations by the fluxgate magnetometer show that there is just a small disturbance of magnetic field during the overturning of electric field. The other way is the electrostatic induction following the Coulomb's law ($\vec{E} = \frac{1}{4\pi\varepsilon_0} \frac{Q}{r^2} \hat{r}$). The connection between the lightning stroke and the overturning of electric field could be explained by a classic thunderstorm charge model through the electric imaging method based on the Coulomb's law (i.e., this model could be supported by a classic electrodynamics textbook written by D.J. Griffiths, 1999). A typical thundercloud (e.g., pairs of ($Q_1$, $-Q_2$) or ($-Q_3$, $Q_4$) in Fig. A1), with a dipole of positive charge located above a negative charge part, would produce an upward electric field toward the ionosphere (see more details in Appendix A).

According to the observations from WWLLN, we find two regions (red ovals A and B in Fig. 4) with heavy lightning activities during the period of the $Na_S$. Before the $Na_S$ occurrences, there were only a few powerful lightnings detected within about (25.1ºN ~ 35.8ºN) and (113.8ºE ~ 118.1ºE) during the period of 12 UT to 13:15 UT (just one strong stroke with a power of 43720.25 kW happening on 12:17 UT, at 25.7229ºN and 117.3955ºE). The continuous strongest lightnings with a power larger than $10^4$ kW occur from 13:19 UT to 15:43 UT, mainly concentrating in two areas centered around (35.8ºN, 118.1ºE) and (25.1ºN, 113.8ºE). After 15:45 UT, no strong strokes were detected again within this area. Thus, the pairs of ($Q_1$, $-Q_2$) and ($-Q_3$, $Q_4$) could be referred to the lightning area of part A and B in Fig. 4. Since thunderstorms could trigger the breakdown process within a rather large area (Leblanc et al., 2008) and influenced the ionosphere with around more than 800 km range horizontally away from the lightning center (Johnson and Davis, 2006; Johnson et al., 1999), the whole area above might undergo a breakdown easily around $Q_1 \sim Q_4$ (e.g., the whole shadow zone in Fig. 4, involving the two strongest lightning areas and the two observing stations).

In previous studies, lightning strokes in lower atmosphere were reported to cause a reduction of electrons of the ionosphere (Shao et al., 2013), and in reverse an enhancement of sodium density in the metal layer (Yu et al., 2017). Those two scenarios are in accord with our current results presented above, with a depletion of $E_S$ and a consequential occurrence of $Na_S$. Although such an idea/picture has been proposed long time ago (Griffiths, 1999), this is the first time that one can apply the imaging method for observing thunderstorms to explain the link between upper and lower atmospheres through an overturning of upward electric field.

Furthermore, the results from different channels of Wuhan ionosonde exhibit extraordinary echoes in different modes during the lightning period (Fig. 5a to 5l). (a) ~ (c): From 13:15 UT to 13:45 UT, the echoes gradually increase. Note that the powerful lightning period begins on 13:15 UT as well, with the sodium density enhancement and the $E_S$ depletion occurring on about 13:20 UT. (d) ~ (g): Most intense echo signals occur during 14:00 UT to 14:45 UT, while the largest intensity of sodium enhancement begins at 14:20 UT and the sodium density peaks at 14:40 UT. The overturning of electric field also occurs at 14:20 UT. (h) ~ (j): From 15:00 UT to 15:30 UT, the signals weaken gradually; (k) ~ (l): The echoes vanish after 15:45 UT. Afterwards, no strong stroke detected again in the discussed area. Meanwhile, the ionospheric echoes diminish

after 15:45 UT, and the overturning of electric field also recovers at about 15:30 UT. Thus, in this case the ionospheric echoes and the lightning activities exhibit an obvious synchronous behavior.

### 3.2 Possible mechanism for $Na_S$

Nominally, the mid-latitude $E_S$ layers would be brought down gradually by tidal fluctuations (Mathews, 1998). The $E_S$ theory predicts that when a series of $E_S$ layers descend below 100 km, metal ions will be depleted through many chemical reactions involving ions and electrons (Cox and Plane, 1998). The main chemical reactions and corresponding rate coefficients for the sodium species under the mesopause condition are summarized in Table 2 (Cox and Plane, 1998; Jiao et al., 2017; Plane et al., 2015; Plane, 2004; Yuan et al., 2019). Application of reaction branching probabilities to reactions 3 to 11 yields the following first-order rate coefficients for the neutralization rates of $Na^+$ ions (Plane, 2004):

$$k(Na^+ \rightarrow Na) = k_3[N_2][M] \times Pr(Na^+ \cdot N_2 \rightarrow Na) + k_4[CO_2][M]$$

$$= k_3[N_2][M] \left( \frac{k_{11}[e^-] + k_5[CO_2]}{k_{11}[e^-] + k_5[CO_2] + k_6[O] \times Pr(NaO^+ \rightarrow Na^+)} \right) + k_4[CO_2][M]$$

$$= k_3[N_2][M] \left( \frac{k_{11}[e^-] + k_5[CO_2]}{k_{11}[e^-] + k_5[CO_2] + k_6[O] \left( \frac{k_7[O] + k_9[O_2]}{k_6[O] + k_8[N_2] + k_9[O_2] + k_{10}[CO_2] + k_{11}[e^-]} \right)} \right) + k_4[CO_2][M]$$

where $Pr$ denotes the branching probability. The first-order conversion rate of $k$ ($Na^+ \rightarrow Na$) can be computed as a function of height using typical values for $N_2$, $O_2$, $O$, $CO_2$ from a WACCM-Na model (Yuan et al., 2019). The results are given in Figure 6. The simulation results show the inflection point of $k$ ($Na^+ \rightarrow Na$) comes out at around 100 km, and below that altitude, the sodium ions would recombine with electrons efficiently through cycling chemical reactions under a large $k$ value. Then the production rate of Na could be obtained from d[Na]/dt = $k$ ($Na^+ \rightarrow Na$)[$Na^+$] (Plane, 2004). The foE$_S$ observed at 14:45 UT equals to 5.75 MHz, indicating a number density of electrons of $e \approx 1.24 \times 10^4 f_o E_S^2 = 4.1 \times 10^5 (cm^{-3})$. If the positive ions are mostly composed of metal ions, with an observed ratio of [$Na^+$] to be $7.41 \times 10^{-2}$ (Kopp, 1997), the estimated [$Na^+$] equivalents to $3.03 \times 10^4 cm^{-3}$. The approximate value of $k$ ($Na^+ \rightarrow Na$) below 100 km is equal to $10^{-4} s^{-1}$. Then the production rate of Na d[Na]/dt = $k$ ($Na^+ \rightarrow Na$)[$Na^+$] equals to 3.03 cm$^{-3}$s$^{-1}$, in accord with the required source strength of sodium atoms of 3 sodium atoms cm$^{-3}$s$^{-1}$ for the formation of $Na_S$ (Cox et al., 1993).

Overall, this $E_S$ mechanism is most widely accepted. Figure 2(b) shows $E_S$ descending near 100 km at about 13:20 UT. Then the $E_S$ depletes, and a moderate enhancement of Na occurs from 13:30 UT to 14:00 UT (shown in Figure 1(a) and 3(a)). This increase in sodium density exhibits no obvious peak, which could probably be in accord with a normally descending $E_S$ governed by tides. In comparison, the peak profile of the $Na_S$ shows intense enhancement and sharp peak, indicating a distinct mechanism.

On the other hand, a link between the reverse electric field and $E_S$ variations could be established through the acceleration of electrons. In classical electromagnetism, positive particles will move along the direction of electric field, and negative particles do opposite (Griffiths, 1999). Since metal ions are much heavier than electrons, the ions would drag electrons to move/drift together; a process known as the bipolar diffusion (Griffiths, 1999). So during the initial phase under a quasi-equilibrium condition, the ions and electrons would co-move downward together along a southward electric field. In a partially ionized plasma, the characteristic frequencies for ions and electrons are associated with the collisions of the plasma particles with stationary neutrals (e.g., the electron–neutral collision frequency $v_{en}$ and the ion–neutral collision frequency $v_{in}$). The collision frequency $v_{sn}$ for scattering of the plasma species $s$ by the neutrals is

$$v_{sn} = n_n \sigma_s^n V_{Ts}, \hspace{2cm} \text{(7) (Shukla and Mamun, 2002)}$$

where $n_n$ is the neutral number density,

$\sigma_s^n$ is the scattering cross section (which is typically of the order of $5 \times 10^{-15}$ cm$^2$ and depends weakly on the temperature $Ts$),

and $V_{Ts} = (k_B T_s/m_s)^{1/2}$ is the thermal speed of the species $s$.

Then under a nonequilibrium phase (e.g., at the point of the electric field overturning), each plasma species has a different relaxation time $\tau = \frac{1}{v}$ (the time needed for reestablishing equilibrium again through collisions). The relaxation time for ions and electrons would be quite different in a partially ionized plasma with the electrons responding much faster than the heavier sodium ions do (since $m_i \gg m_e$). This discrepancy would cause a charge separation temporarily. The single electrons move opposite along the electric field, which means during the upward electric field they would be rapidly accelerated downward, while the ions could be regarded as essentially remaining unchanged. The electrons would reverse rapidly before the ions can respond similar to the velocity overshoot effect for electrons. During the relaxation phase, the recombination between electrons and ions would probably be triggered through collisions, not unlike how moving cars will crash in a traffic accident if the car behind suddenly accelerates.

Based on the above results, a possible mechanism for $Na_S$ could be suggested by the following four steps: (1) Strong lightning strokes produce an upward atmospheric electric field toward the ionosphere; (2) The reverse of electric field would cause a temporary charge separation, leading to a trigger of recombination between electrons and ions; (3) When the $E_S$ descends below about 100 km, the sodium ions would recombine with electrons much more efficiently through cycling chemical reactions under a large $k$ (Na$^+$→Na) value; (4) The depleted $E_S$ layers generate the formation of $Na_S$. Thus, we propose that there would probably be a connection between the lightning strokes, overturning of the electric field,

ionospheric disturbances, and also the Na$_S$. A link between the lower and upper atmospheres could be established by carefully studying and examining these phenomena. However, we caution that the key processes for our proposed step (2) remained still quite uncertain. A more in-depth modelling study concerning both plasma and neutral molecules is needed in the future.

## 4 Conclusions

In this research, we study the conjunction between the lower and upper atmospheres, through the phenomena and processes of lightning strokes, overturning of the atmospheric electric field, ionospheric disturbance, plasma drift velocity reversal, and the formation and dissipation of sporadic sodium layer. The main findings of our results are summarized as follows:

1. The Na$_S$ event discussed in the present case study shows a close relationship with E$_S$ activities rather than conforming with the prescriptions from the competing high temperature theory.

2. The atmospheric electric field exhibits an overturning, opposite to the fair-weather downward field in the global circuit, in coincident with the depletion of E$_S$ and the consequent production of Na$_S$.

3. A statistical analysis shows that the E$_S$ could hardly be formed or maintained when the atmospheric electric field is directed upward.

4. A typical thunderstorm, with a positive charge located above a negative charge layer, is shown to produce an upward electric field toward the ionosphere. Two regions with heavy lightning activities nearby are found during the overturning of the atmospheric electric field.

5. Observations by the ionosonde exhibit extraordinary echoes during the lightning period and the temporal property of the echoes behaved synchronously with lightning activities.

6. WACCM-Na model simulation results show that the calculated first-order rate coefficient $k$ (Na$^+\rightarrow$Na) could probably explain the efficient recombination of Na$^+\rightarrow$Na in this Na$_S$ case study.

Our results support a physical connection between the lightning strokes, overturning of the electric field, ionospheric disturbances, and possibly the Na$_S$ phenomenon as well. A link between the lower and upper atmospheres could be established by the monitoring of Na$_S$ and related phenomena as follows: lightning strokes$\rightarrow$overturning of electric field$\rightarrow$depletion of E$_S$/generation of Na$_S$.

## Appendix A: Calculations for the induced upward electric field in the global electric circuit

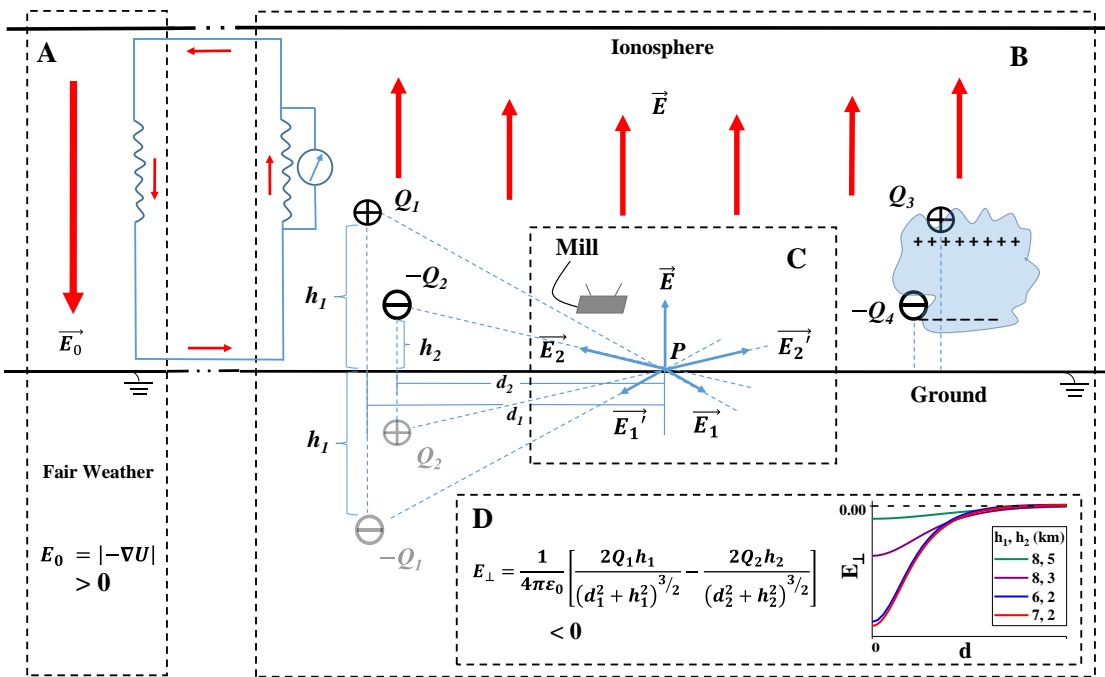

**Figure A1: A diagram illustrates the global electric circuit. Part A: The atmospheric electric field under fair weather with a downward field returning from the ionosphere. Part B: The dynamo area, with thunderstorms generating an upward electric field towards the ionosphere. The electric field intensity $E_\perp$ could be deduced through the electric imaging method. Part C: The deduced vertical electric field intensity at any point $P$ within the thundercloud. Part D: The calculated $E_\perp$ based on the electric imaging method.**

Suppose there is positive charge $Q_1$ at the top of a thunderstorm, with a distance of $d_1$ above the ground; and a negative charge—$Q_2$ at the bottom with a distance of $d_2$. Since the ground surface could be regarded as an infinite conducting plane, it would generate an induced charge. The boundary conditions here is:

$$U = 0 \quad \text{at } z=0$$

$$U \rightarrow 0 \quad \text{at infinity}$$

Under the uniqueness theorem, we can remove the ground surface if we put the postulated image charges of—$Q_1$ and $Q_2$ to the corresponding mirror points. Then for an arbitrary point $P$ near the boundary, the vertical electric field equals to the following expression according to the Coulomb's law:

$$E_\perp = \frac{1}{4\pi\varepsilon_0}\left[\frac{2Q_1 h_1}{\left(d_1^2+h_1^2\right)^{3/2}} - \frac{2Q_2 h_2}{\left(d_2^2+h_2^2\right)^{3/2}}\right]. \qquad \text{(A1) (Griffiths, 1999)}$$

In the simplest case, when $Q_1$ equals to $Q_2$ and $d_1=d_2=d$, $E_\perp$ varies with the distance $d$. If $Q_2$ is larger than $Q_1$ ($Q_2 > Q_1 > 0$), and the negative charge—$Q_2$ is more closed to the observing point P ($d_2 < d_1$), $E_\perp$ would acquire negative values (e.g., with the upward direction). A brief simulation result is shown by part D, exhibiting a persistent negative values for $E_\perp$.

**Data availability**

The data sets of sodium fluorescence lidar at Hefei and three kinds of instruments at Wuhan (the ionosonde, electric mill and the fluxgate magnetometer) are publicly available from the Chinese Meridian Project database at http://data.meridianproject.ac.cn/. The access to the sodium density and temperature data by the USTC T/W lidar is referred to National Space Science Data Center, National Science & Technology Infrastructure of China (http://www.nssdc.ac.cn). The lightning location and power data can be downloaded from the World- Wide Lightning Location Network (http://wwlln.net/).

**Acknowledgements**

This work is supported by the National Natural Science Foundation of China (41974178) and the National Key R&D Program of China (2017YFC0602202). We acknowledge the use of data from the Chinese Meridian Project for the sodium fluorescence lidar at Hefei and three kinds of instruments at Wuhan (the ionosonde, electric mill and the fluxgate magnetometer, http://data.meridianproject.ac.cn/). We thank the World-Wide Lightning Location Network (http://wwlln.net/), a collaboration among over 50 universities and institutions, for providing the lightning location data used in the plotted figures. We thank Dr. Wuhu Feng for his help on model simulations. We also acknowledge the constructive reviews by both referees. The first author would like to thank Chen Hao and Liu Yandong for their help on debugging programs.

**Author information**

**Affiliations**

**Department of Geophysics, College of the Geology Engineering and Geomatics, Chang'an University, Xi'an, 710054, China**

Shican Qiu & Ning Wang,

**Key Laboratory of Geospace Environment, Chinese Academy of Sciences, University of Science & Technology of China, Hefei, Anhui, 230026**

Shican Qiu, Gaopeng Lu, Mingjiao Jia, Xingjin Wang, Xianghui Xue, Tao li & Xiankang Dou

**Mengcheng National Geophysical Observatory, School of Earth and Space Sciences, University of Science and Technology of China, Hefei, Anhui, 230026, China**

Mingjiao Jia, Xingjin Wang, Xianghui Xue, Tao li & Xiankang Dou

**Gravity & Magnetic Institute of Chang'an University, Xi'an, 710054, China**

Ning Wang

**Key Laboratory of Western China's Mineral Resources and Geological Engineering, China Ministry of Education, Xi'an, 710054, China**

Ning Wang

**Center for Environmental Research and Earth Sciences (CERES), Salem, Massachusetts, 01970, USA**

**and Institute of Earth Physics and Space Science (ELKH EPSS), H-9400, Sopron, Hungary**

Willie Soon

## Contributions

Shican Qiu conceived this study and wrote this manuscript. She also prepared Fig. 2~5 in the main text and Fig. A1 in the Appendix.

Ning Wang performed data analysis.

Willie Soon was in charge of the organization and English polishing of the whole manuscript.

Gaopeng Lu added some materials about thunderstorms and lightning strokes.

Mingjiao Jia prepared Fig. 1 and gave some useful comments on the content.

Xingjin Wang wrote the response to reviewers and calculated the model simulation results.

Xianghui Xue wrote the response to reviewers and added some materials in the discussion.

Tao Li helped with the response to reviewers.

Xiankang Dou conceived this study and provided data from the Chinese Meridian Project.

## Competing interests

The authors declare no conflict of interest.

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

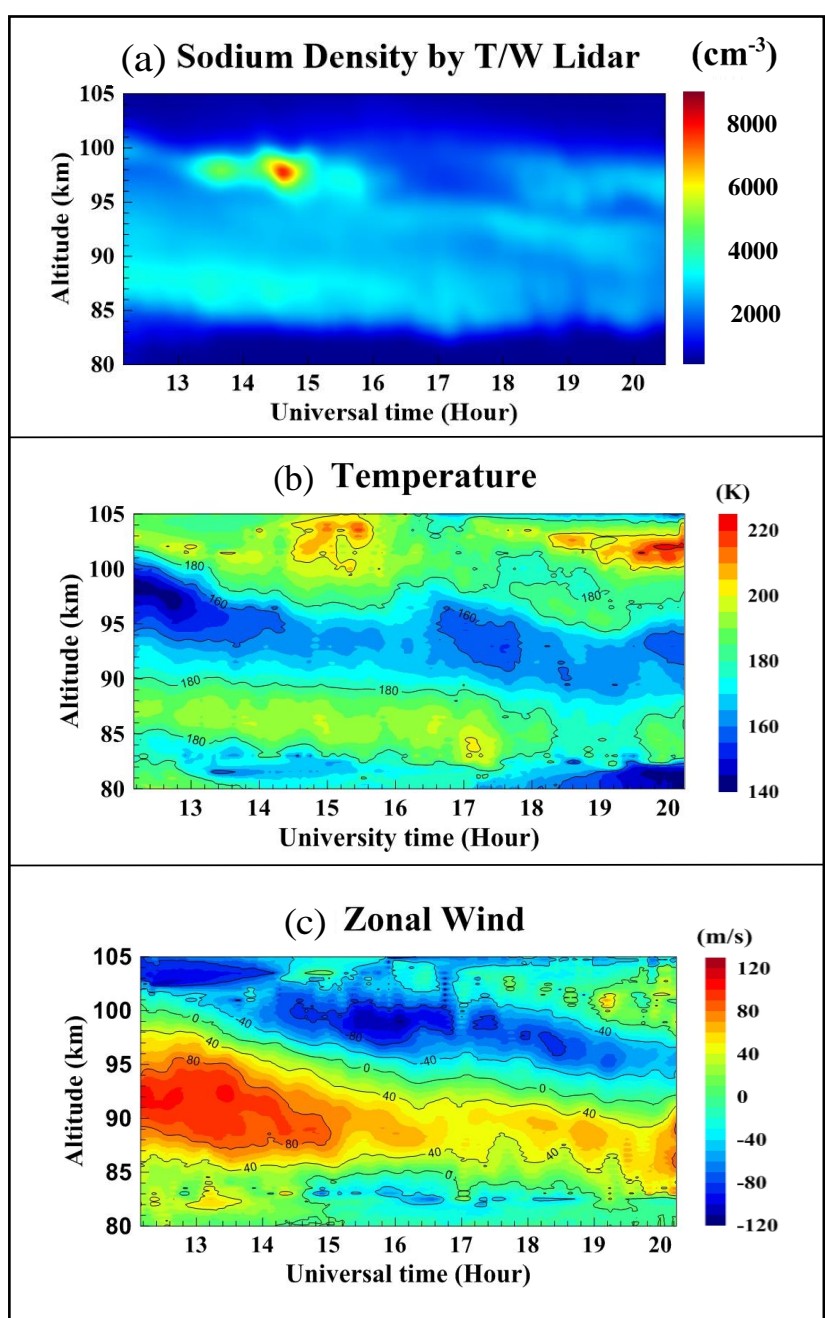

**Figure 1: Observations on June 3rd, 2013, by the USTC T/W lidar. (a) The sodium density profile of the west beam by T/W lidar A moderate increase of sodium density appears at about 13:20 UT, while the largest intensity of sodium enhancement begins at about 14:20 UT. The sodium density peaks at 14:37 UT around 97.65 km. (b) Temperature profile observed by the T/W lidar, showing a cold region where the Na$_S$ occurs. (c)The zonal wind detected by the T/W lidar, exhibiting a suitable wind shear for the creation or formation of E$_S$.**

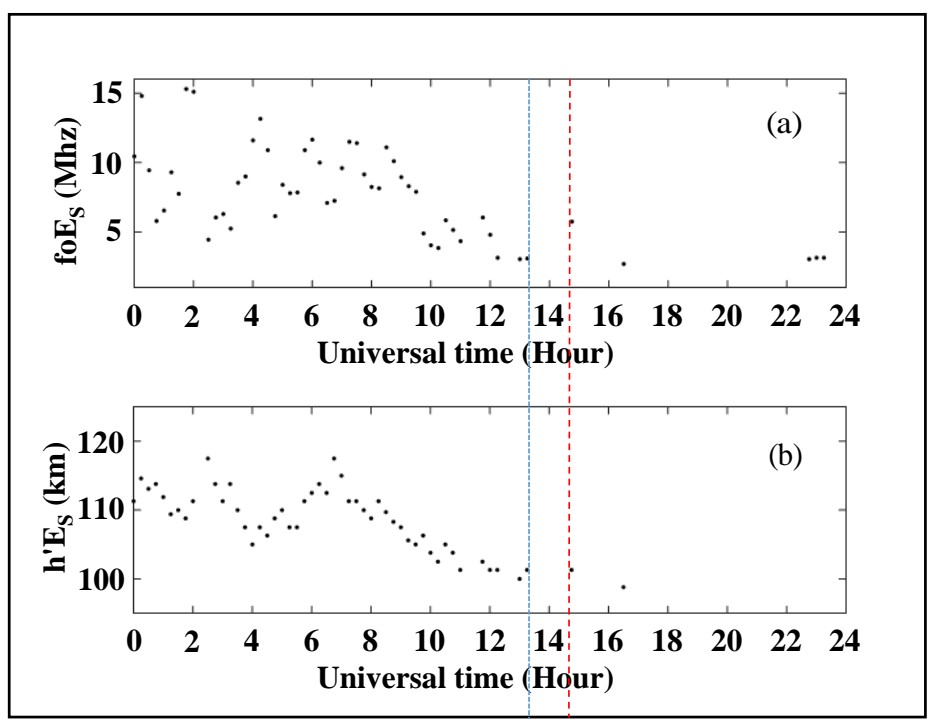

**Figure 2: Sporadic E layers observed by the ionosonde at Wuhan (30.5°N,114.6°E). (a) The time series of the critical frequency for E$_S$ (foE$_S$). The E$_S$ layers travel/propagate downward starting around 6:30 UT, and deplete altogether at about 13:20 UT. (b) The visual height of E$_S$ (h'E$_S$). The vertical blue dotted line annotates the beginning of the Na$_S$ around 13:20 UT, and the vertical red dashed line points out the time when the most intense sodium enhancement starts on 14:20 UT.**

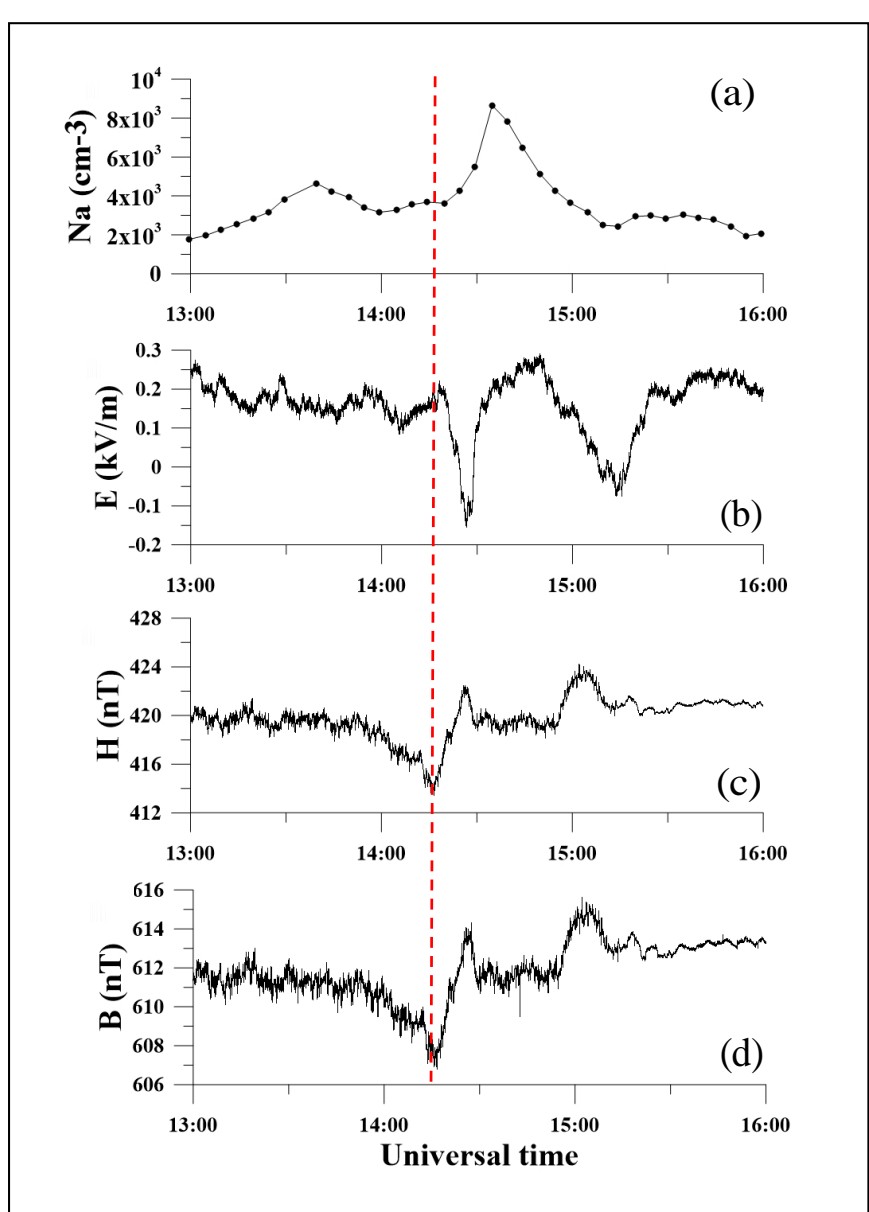

**Figure 3: Observations of some atmospheric parameters and some deduced results. (a) Time series of sodium density variations at peak height 97.65 km, observed by the west beam of the T/W lidar. The sodium density begins to increase at about 14:20 UT. (b) Atmospheric electric field variations, exhibiting a synchronous overturning from 14:20 UT with the enhancement of sodium density (also pointed out by the vertical red dashed line). Note that there is another overturning peaking at 15:15 UT, without another $Na_S$ being produced, which could be explained by a depletion of ions in the $E_S$. The electric field recovers at about 15:30 UT. (c) Horizontal magnetic field observed by the fluxgate magnetometer. (d) The deduced magnetic induction intensity from observations of $(H, D, Z)$ components by the fluxgate magnetometer ($B = \sqrt{H^2 + D^2 + Z^2}$).**

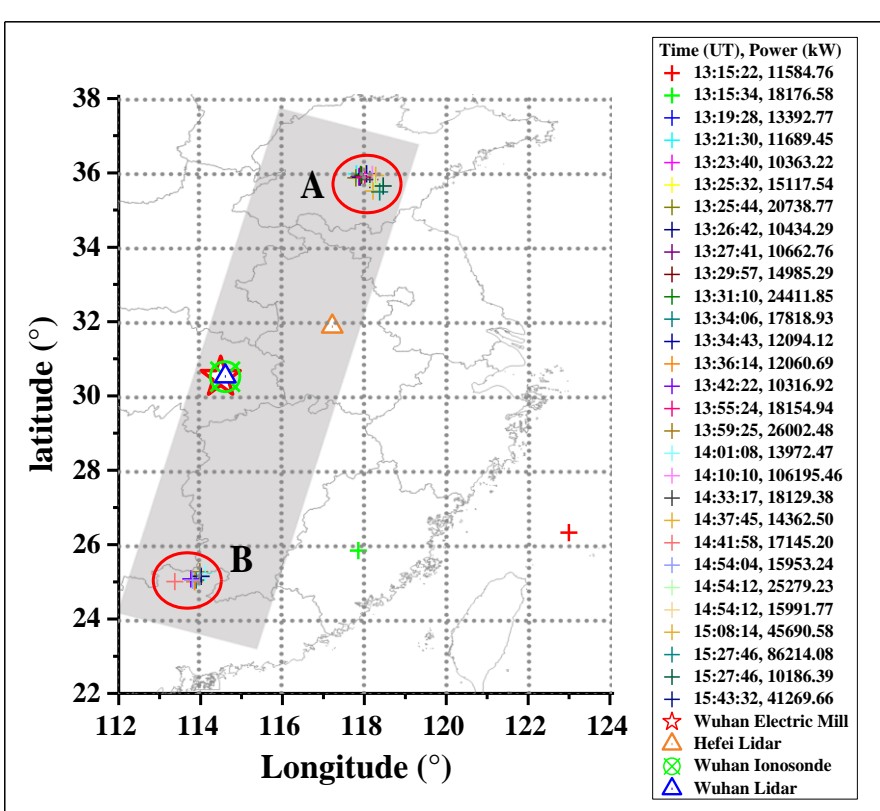

**Figure 4: The lightning strokes are detected by WWLLN. The continuous strongest lightnings with a power larger than $10^4$ kW occur from 13:19 UT to 15:43 UT, mainly concentrating around areas within (35.8ºN, 118.1ºE) and (25.1ºN, 113.8ºE), respectively.**

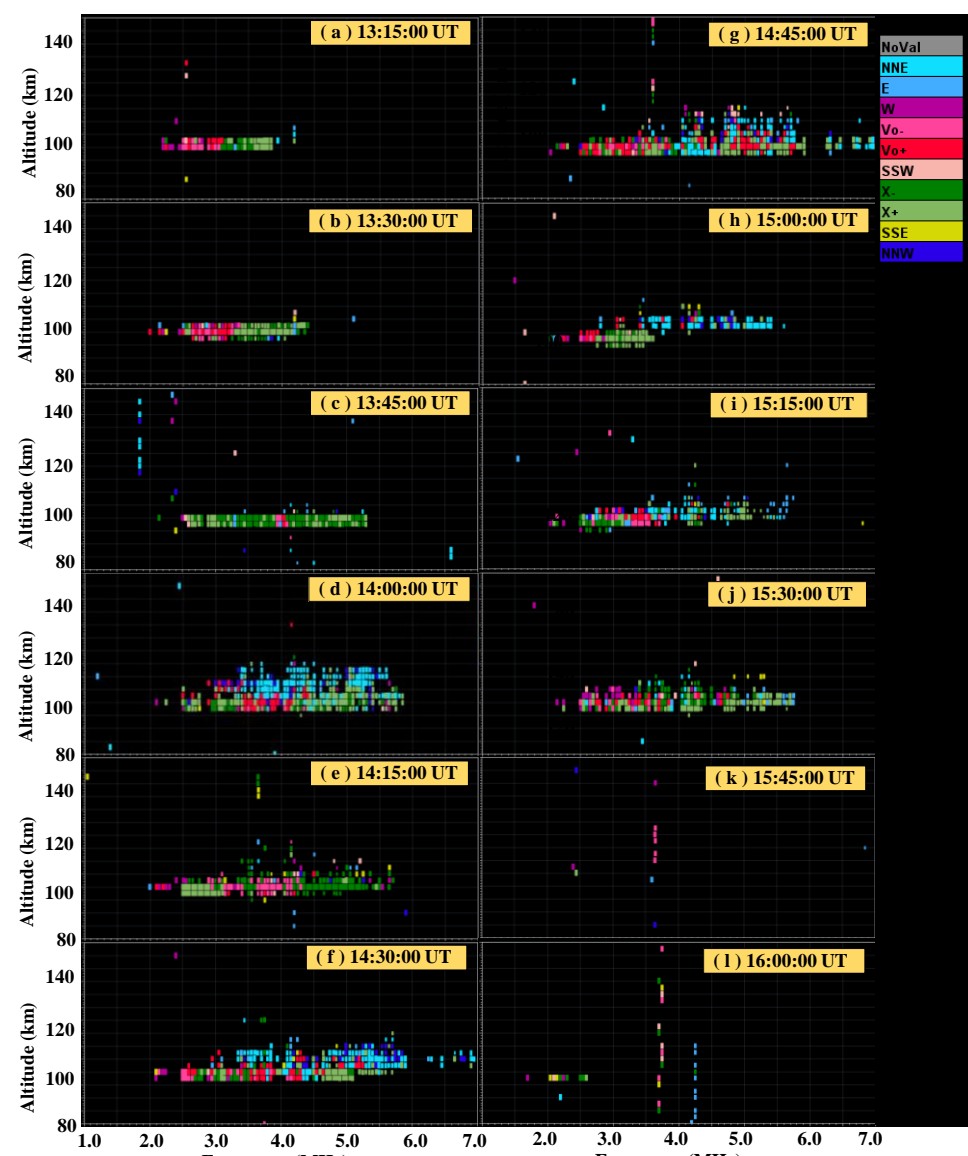

**Figure 5: Extraordinary echoes by Wuhan ionosonde in different modes: (a) ~ (c): From 13:15 UT to 13:45 UT, the echoes gradually increase. Note that the powerful lightning period begins on 13:15 UT as well, with the sodium density enhancement and the Es depletion occurring on about 13:20 UT. (d) ~ (g): Most intense echo signals occur during 14:00 UT to 14:45 UT, while the largest intensity of sodium enhancement begins at 14:20 UT and the sodium density peaks at 14:40 UT. The overturning of electric field also occurs at 14:20 UT. (h) ~ (j): From 15:00 UT to 15:30 UT, the signals weaken gradually. (k) ~ (l): The echoes vanish after 15:45 UT. Afterwards, no strong stroke detected again within the discussed area. Meanwhile, the ionospheric echoes diminish after 15:45 UT, and the overturning of electric field also recovers at about 15:30 UT.**

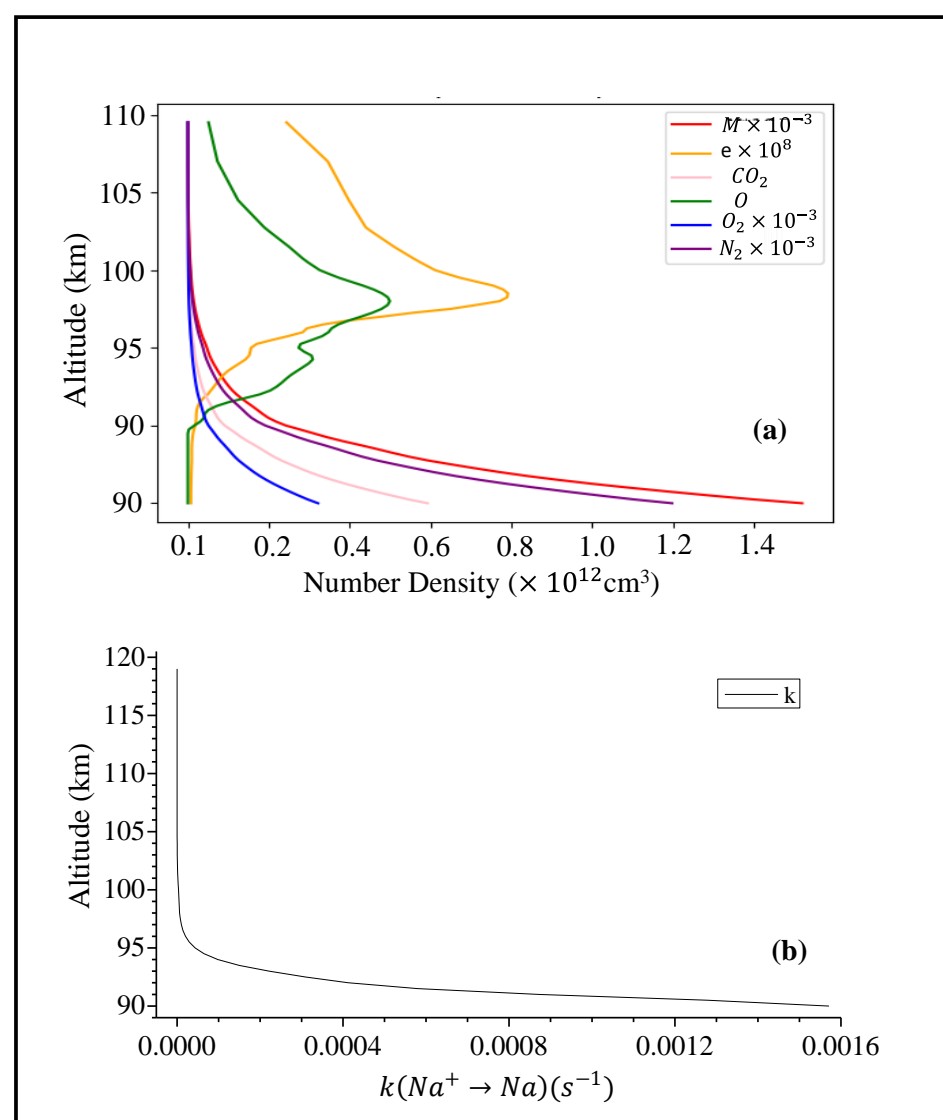

**Figure 6: WACCM-Na Model simulation results. (a)** Constituents of the species used for calculating $k$ (Na$^+\rightarrow$Na). The number

densities of CO₂, O₂, O, the total atmosphere density M and $N_2 \approx [M] - [O_2] - [O]$ are derived from Yuan et al., 2019. The

number density of electrons equals to $e \approx 1.24 \times 10^4 f_o E_s^2 \left(cm^{-3}\right)$. **(b)** The calculated first-order rate coefficient

5    $k$ (Na$^+\rightarrow$Na), indicating much more efficient recombination below about 95-100 km.

**Table 1 : The nature of foE$_S$ variations under the situations of the overturning of electric field.**

| foE$_S$ Variations | Termination | Decrease | Generation | Increase | Others |
|---|---|---|---|---|---|
| **Ratios** (Case days/Total days) | 155/242 | 39/242 | 6/242 | 26/242 | 28/242 |

**Table 2: Ion-molecule reaction rate coefficients for Na summarized from previous reports (Cox and Plane, 1998; Jiao et al., 2017; Plane et al., 2015; Plane, 2004; Yuan et al., 2019).**

| No. | Reaction | Rate Coefficient[a] |
|---|---|---|
| 1 | $Na + O_2^+ \rightarrow Na^+ + O_2$ | $2.7 \times 10^{-9}$ |
| 2 | $Na + NO^+ \rightarrow Na^+ + NO$ | $8.0 \times 10^{-10}$ |
| 3 | $Na^+ + N_2(+M = N_2 \,\&\, O_2) \rightarrow Na^+ \cdot N_2$ | $4.8 \times 10^{-30}(T\,/\,200K)^{-2.2}$ |
| 4 | $Na^+ + CO_2(+M) \rightarrow Na^+ \cdot CO_2$ | $3.7 \times 10^{-29}(T\,/\,200K)^{-2.84}$ |
| 5 | $Na^+ \cdot N_2 + CO_2 \rightarrow Na^+ \cdot CO_2 + N_2$ | $6.0 \times 10^{-10}$ |
| 6 | $Na^+ \cdot N_2 + O \rightarrow NaO^+ + N_2$ | $4.0 \times 10^{-10}$ |
| 7 | $NaO^+ + O \rightarrow Na^+ + O_2$ | $1.0 \times 10^{-11}$ |
| 8 | $NaO^+ + N_2 \rightarrow Na^+ \cdot N_2 + O$ | $1.0 \times 10^{-12}$ |
| 9 | $NaO^+ + O_2 \rightarrow Na^+ + O_3$ | $5.0 \times 10^{-12}$ |
| 10 | $NaO^+ + CO_2 \rightarrow Na^+ \cdot CO_2 + O$ | $6.0 \times 10^{-10}$ |
| 11 | $Na^+ \cdot X\ (X = O, N_2, CO_2) + e^- \rightarrow Na + X$ | $1.0 \times 10^{-6}(200K\,/\,T)^{0.5}$ |

[a] Units: bimolecular reactions, cm$^3$ molecule$^{-1}$ s$^{-1}$; termolecular reactions, cm$^6$ molecule$^{-2}$ s$^{-1}$