# Peer review of "Sporadic sodium layer: A possible tracer for the conjunction between the upper and lower atmospheres"

_Atmospheric Chemistry and Physics, 2020_

## Referee Comment (RC1) · Anonymous Referee #2 · 25 Jan 2021

**General comments:**

A reasonable mechanism of Nas layer enhancement in MLT region by lower atmospheric electric field was proposed in this manuscript, based on the observation date of several kinds of detection tools. This paper also provided a detail process how lower atmospheric electric field influences the ionized and neutral components in the upper atmosphere. This research is so novel and the similar reports are quite little up towards now. Consequently this manuscript fitted the scope of ACP at the moment in my point of view, however the mechanism of Nas layer enhancement in this manuscript needs to be further analyzed in the proceeding steps.

**Specific comments:**

1. In Fig.1a, the peak density of Nas is more than 12000 $cm^{-3}$ on 97.75 km at 14:40 UT (line 142). But it can be seen from Fig.1b, the peak density of Nas at the same time is only about 5000 $cm^{-3}$. Although the sodium density in Fig.1a and Fig.1b were given by two different kinds of lidars, the large difference of the density is difficult to be understood as these two lidars almost located at the same site. Is this large difference caused by the different resolutions or reversion methods? Authors should re-calculate the density in Fig.1a and Fig.1b, and give reasonable values.

2. In the Discussions, a clear causal chain is given by the authors: the lightning strokes induced the overturning of the electric field, and then induced the ionospheric disturbances, as well as Nas. However, it can be seen from Fig.2a: there is an enhancement in the Es layer from 13:20 to 14:20, and the origin of this enhancement was not explained or discussed in the manuscript. Is it caused by lightning as proposed by Johnson and  Davis (GRL, 2006) ? As the main contribution of this manuscript is to propose a new mechanism of Nas layer enhancement by lower atmospheric electric field, I suggest that Authors could explain or discuss the enhancement in the Es layer from 13:20 to 14:20.

3. From the progress of Nas in Fig.1a:The density of Nas increases with height drops after the Nas height lower than 100 km. A maximum of Nas is present at 97km around 13:30, and later the density of Nas decreases. But another maximum of Nas is present again around 14:20 (almost the same height), and the authors think that this Nas is produced by the overturn of low-altitude electric field.

According to Plane's theory, when the Es drops below 100 km, Na ions in the Es are rapidly neutralized to form Nas. As the height decreases, the rate of Na ion neutralization increases. And so there's Nas maximum at about 13:20 UT around 97 km. Later, the Nas gradually weakened due

to excessive consumption of Na ions. But for the bigger Nas around 14:20, there needs to be an implicit condition if the ions in the Es contribute. That is, Na ions in the Es increased at 13:30. This may be due to the addition of surrounding sodium ions resulting in Es increase. So there may be a possible mechanism: A reversal of the electric field adds sodium ions nearby, and these ions enhance Es. And then the sodium ions in the Es are neutralized to form Na atoms, with Es weakened.

**Technical corrections:**

1. From line 22 to line 26 in the abstract: rewrite this long sentence.

2. From line 47 to line 49, "the metal layers (especially the sodium layer), which located between about 80~110 km, could possibly act as a window to detect the MLT parameters by means of fluorescence resonance lidars." please add the corresponding reference.

3. And from line 49 to line 51 please also add the reference.

4. At the end of Line 64, please consider the word "candidate" whether is proper or not.

5. The author should pay much attention to the tense of the manuscript,the past tense shall be used when giving background information in the Abstract and Introduction, when describing methods used, and when presenting and discussing results. There

are indeed quite a lot of such serious problems throughout all the manuscript. If possible, please ask a native speaker for help.

6. Line 137, please consider about the abbreviation "T/W lidar" as it appeared there for the first time.

7. Please change the line 162 as "in accord with our previous reports WHICH shown that an Nas higher than 96 km tended to be…

8. Line 182, Based on the ABOVE observations

9. Line 215 also POINTED out by the vertical red dashed line.

10. Line 234 could be changed to "It is worth mentioned that…"

11. Line 254 could be changed to "The electric field could change through within two distinct ways as below:"

12. Line 291-292: "mainly concentrating in two ranges about (35.8ºN, 118.1ºE) and (25.1ºN, 113.8ºE)." please rewrite this sentence.

13. Line 302 could be changed to "Afterwards, no strong stroke WAS detected again in the discussed area. "

14. Line 311-313 the caption of Figure 5 (a)~(l): please unify the tense of verbs and pay attention to the English writing again as I remind.

15. Lines 335-336, "similar to how moving cars will crash in a traffic accident in the car in front suddenly turns back or brakes" Please rewrite this sentence within much more scientific aspect.

16.Line 338, are you sure by three steps? Please check it.

---

## Referee Comment (RC2) · Anonymous Referee #1 · 29 Jan 2021

This paper describes a study of sporadic sodium layers (Nas) and their possible relationship with strong lightning discharges causing the overturning of the electric field in the upper atmosphere. An impressive range of instruments was used for the study: two Na lidars in Hefei, one of which also measured wind and temperature; a third Na lidar in Wuhan; and an ionosonde and electric field mill in Wuhan. An Nas layer was observed by the Hefei lidars (though not by the Wuhan lidar?), which coincided with an overturning of the vertical electric field. Strong lightning was observed in the region, and the authors postulate that highly charged clouds led to the overturning of the electric field, and that this may be causally linked to the appearance of the Nas.

While the possible link with the overturned electric field and lightning is an interesting idea and certainly worth investigating, I am not sure that a link with the overturned elec-

tric field is needed to explain the observed Nas. Figure 1(d) shows a very clear shear in the zonal wind, which descends below 100 km around 1330 hrs. This coincides with the observed descent of the Es layer to 100 km (Figure 2b), after which it disappears and the Nas appears (Figure 1a and 1b). The Na density in the Nas layer peaks around 1.2e4 cm-3. The strength of the Es layer as it descends to 100 km is around 4 MHz, corresponding to an electron density of 2e5 cm-3. Most of the metallic ions in the Es layer would be Fe+ and Mg+, with a smaller amount of Na+. Assume all the Na+ ions were neutralised when the Es layer descended below 100 km and the ion-molecule chemistry becomes very fast (the theory proposed quantitatively by Plane). Then, if you divide the peak Na density in the Nas layer by the Es electron density, this implies that the fraction of Na+ in the Es layer was around 6%, which sounds sensible (see the results of rocket-borne mass spectrometers flown by E. Kopp, for example).

What this exercise shows is that the Nas can be explained by the sporadic E layer descending below 100 km. The authors therefore need to explain what the additional effect of the overturned electric field might be. If it is not needed to explain the appearance of the Na layer, then the two phenomena could be quite unrelated to each other.

In fact, the statistics summarized in Table 1 indicate that the overturned electric field is often associated with the termination or significant decrease in Es layers (80% of the time). So the question is what is the link? Does the reversal of the electric field accelerate the Es downwards, leading to its destruction through fast ion-molecule chemistry and the appearance of Nas?

Other matters to address:

1. The Na lidar measurements in Figure 1(a) and 1(b) have concentrations varying by a factor of 2, even though co-located in Hefei. This cannot be correct. Even if the lidars have different vertical resolutions, the integrated Na density across the Nas layer should be the same.

2. The statement on page 6 (line 28): "Thus, we conclude that the ionospheric echoes and the lightning activities exhibit an obvious synchronous behaviour." Presumably Es layers are observed over Wuhan in the absence of lightning activity. So the implication of this statement is that whenever strong lightning is present Es is observed – is that correct?

Minor issues:

page 2, line 4: I don't think the MLT is the "least known part of our planet" – what about the deep oceans? I think you mean "our planet's atmosphere".

page 2, line 6: changed "sodium species" to "sodium atoms", since that is the form of Na that can be observed from the ground.

page 2, lines 15-30: in this discussion there is no mention of the magnetic field, which is part of the classical VxB mechanism for sporadic E formation

page 4, line 10: "prefers the Es mechanism" implies that the Nas is an intelligent being that can make a choice! I would rephrase "is better explained by the Es mechanism"

page 4, line 16: "suggests"

page 5, line 4: provide a citation for the equation

page 5,line 29: the phrase "could be supported by a classic electrodynamics textbook" should be omitted. Either this is very well understood by the community, or you should provide reference to such a textbook.

page 6, line 6: I cannot believe that the energy of this lightning stroke is known to 7 significant figures!

page 6, line 12: the shading in the figure is too faint to see.

page 6, line 17: the statement "such an idea/picture has been proposed long time ago" must be referenced.

page 6, line 30: the statement "The Sodium ions and electrons recombine much faster during the overturning period, leading to a depletion of the Es" needs further discussion. As I state above, I think this can only happen if the Es descends rapidly. Another possibility is that somehow the Es becomes compressed so that the plasma concentration increases – that seems unlikely.

page 7, line 12: change "consequential occurrence" to something like "consequent production"

page 9, line 5: the journal requires that the data is archived and accessible to the reader. Not through writing to someone.

page 19. Figure Caption 3: change "another Nas companying" to "another Nas being produced" Figure 4 should be redrawn. The site of the electric mill is hidden. The units of power are W, not J, and these "powers" contain too many significant figures. The map and shading are too faint to read.

Figure 5. The legend in the figure needs to be explained in the figure caption.

---

## Author Comment (AC1) · 15 Mar 2021

We would like to thank the reviewer for the valuable comments and constructive suggestions. We have studied all comments carefully and revised the manuscript accordingly. We marked all the changes in red fonts in the revised manuscript. The point-by-point answers to the comments are given below in blue fonts.

This paper describes a study of sporadic sodium layers (Nas) and their possible relationship with strong lightning discharges causing the overturning of the electric field in the upper atmosphere. An impressive range of instruments was used for the study: two Na lidars in Hefei, one of which also measured wind and temperature; a third Na lidar in Wuhan; and an ionosonde and electric field mill in Wuhan. An Nas layer was observed by the Hefei lidars (though not by the Wuhan lidar?), which coincided with an overturning of the vertical electric field. Strong lightning was observed in the region, and the authors postulate that highly charged clouds led to the overturning of the electric field, and that this may be causally linked to the appearance of the Nas. While the possible link with the overturned electric field and lightning is an interesting idea and certainly worth investigating, I am not sure that a link with the overturned electric field is needed to explain the observed Nas. Figure 1(d) shows a very clear shear in the zonal wind, which descends below 100 km around 1330 hrs. This coincides with the observed descent of the Es layer to 100 km (Figure 2b), after which it disappears and the Nas appears (Figure 1a and 1b). The Na density in the Nas layer peaks around 1.2e4 cm$^{-3}$. The strength of the Es layer as it descends to 100 km is around 4 MHz, corresponding to an electron density of 2e5 cm$^{-3}$. Most of the metallic ions in the Es layer would be $Fe^+$ and $Mg^+$, with a smaller amount of $Na^+$. Assume all the $Na^+$ ions were neutralised when the Es layer descended below 100 km and the ion-molecule chemistry becomes very fast (the theory proposed quantitatively by Plane). Then, if you divide the peak Na density in the Nas layer by the Es electron density, this implies that the fraction of Na+ in the Es layer was around 6%, which sounds sensible (see the results of rocket-borne mass spectrometers flown by E. Kopp, for example). What this exercise shows is that the Nas can be explained by the sporadic E layer descending below 100 km. The authors therefore need to explain what the additional effect of the overturned electric field might be. If it is not needed to explain the appearance of the Na layer, then the two phenomena could be quite unrelated to each other.

In fact, the statistics summarized in Table 1 indicate that the overturned electric field is often associated with the termination or significant decrease in Es layers (80% of the time). So the question is what is the link? Does the reversal of the electric field accelerate the Es downwards, leading to its destruction through fast ion-molecule chemistry and the appearance of Nas?

Thanks for the important comment. First, we would like to explain the locations of lidars. The two stations at Hefei and Wuhan locate about 350 km apart. Unfortunately, there was no observations by Wuhan lidar on June 3$^{rd}$, otherwise the results would be far more convincing. However, previous statistical analyses show that Na$_S$ often

observed simultaneously at both stations: "Among all 19 cases, 16 cases, including 9 SSL cases and 7 TeSL cases, occurred almost simultaneously over Hefei and Wuhan without a time delay. Seven TeSLs and four out of the nine SSLs were accompanied by ionospheric sporadic $E$ ($E$s)" (see Ma et al., 2019). So we could possibly assume and infer that the horizontal scale of $Na_S$ is large enough.

Second, we would like to discuss the possible link between electric field and $E_S$, as suggested by the kind reviewer. Since the recombination of $Na^+ + e^- \rightarrow Na + hv$ is inefficient to generate $Na_S$, $Na^+$ ion is believed to first form a ligand $Na^+ \cdot N_2$ through the recombination reaction:

$$Na^+ + N_2 + M \rightarrow Na^+ \cdot N_2 + M, \qquad (1)*$$

with a rate coefficient of $k_1 = 4.8 \times 10^{-30}(T/200)^{-2.2}$ cm$^6$ molecule$^{-2}$ s$^{-1}$ (Cox and Plane, 1998). $Na^+ \cdot N_2$ can either switch with $CO_2$ (which will undergo dissociative electron recombination to form $Na$), or $O$ (which reforms $Na^+$)(Cox and Plane, 1998). The key factor of Professor Plane's $E_S$ mechanism depends on the ratio of [O]/ [$CO_2$]. Recombination of $Na^+ \cdot CO_2$ and $e^-$ will increase rapidly as [O]/ [$CO_2$] decreases below the value of 100 (Cox and Plane, 1998). Then the sodium atoms could be formed directly from the following chemical reaction:

$$Na^+ \cdot CO_2 + e^- \rightarrow Na + CO_2. \qquad (2)* \text{ (Cox and Plane, 1998)}$$

The chemical reaction rate ($v$) for this second-order reaction could be calculated using the following equation:

$$v = k[Na^+ \cdot CO_2]N_e, \qquad (3)*$$

The reaction rate coefficient $k_2$ for the chemical reaction is experimentally measured to be:

$$k_2 = 1 \times 10^{-6}\sqrt{\frac{200}{T}} \quad \text{(cm}^6 \text{ molecule}^{-2} \text{ s}^{-1}) \qquad (4)*$$

(Collins et al., 2002; Cox and Plane, 1998; Daire et al., 2002), and the electron density $N_e$ can be calculated using the following equation:

$$N_e = 1.24 \times 10^4 foE_S^2 (cm^{-3}) \qquad (5)* \text{ (Bittencourt, 2004).}$$

Overall, this $E_S$ mechanism is most widely accepted, if we neglect $k_1$ as being too small with an order of $10^{-30}$. A possible adaptation is to assume a plenty quantity

of pre-existing $Na^+ \cdot N_2$ / $Na^+ \cdot CO_2$ in the sodium layer, and the $E_S$ just needs to provide enough additional electrons.

On the other hand, a link between the reverse electric field and $E_S$ variations could be established through the acceleration of electrons. Normally, positive particles will move along the direction of electric field, and negative particles do the opposite (Griffiths, 1999). Since metal ions are much heavier than electrons, the ions would drag electrons in order to move/drift together, this process is called the bipolar diffusion (Griffiths, 1999). In the initial stage, ions and electrons descend gradually under the southward electric field (Fig. 1*(a)). In a partially ionized plasma, the characteristic frequencies for ions and electrons are associated with the collisions of the plasma particles with stationary neutrals (e.g., the electron–neutral collision frequency $v_{en}$ and the ion–neutral collision frequency $v_{in}$). The collision frequency $v_{sn}$ for scattering of the plasma species $s$ by the neutrals is given by

$$v_{sn} = n_n \sigma_s^n V_{Ts}, \qquad (6)* \text{ (Shukla and Mamun, 2002)}$$

where $n_n$ is the neutral number density,

$\sigma_s^n$ is the scattering cross section (which is typically of the order of $5 \times 10^{-15}$ cm$^2$ and depends weakly on the temperature $Ts$),

and $V_{Ts} = (k_B T_s/m_s)^{1/2}$ is the thermal speed of the species $s$.

So the relaxation times $\tau = \frac{1}{v}$ for ions and electrons are different in a partially ionized plasma, and electrons would respond much faster than the heavier sodium ions do (since $m_i \gg m_e$). At the moment when the electric field reverses, electrons will be rapidly accelerated by the northward electric field, and ions would be regarded as essentially remaining northward or unchanged. If we could possibly assume that Na$^+$·CO$_2$ is always excessive (we see can below it just needs a number density of 100 cm$^{-3}$, but regrettably there is no direct measurements and detections even up till now), we only need to consider the amount of electrons. When the reactant concentration for R(2)* is increased (e.g., when a concentrated electron layer accelerated downward below 100 km), this reaction will shift to the right side of R(2)*.

[Figure]

**Fig. 1\* A sketch to illustrate what happened before (a) and after (b) the overturning of atmospheric electric field: (a) The ions and electrons descend gradually under the southward electric field. (b) At the moment when the electric field reverses, the electrons will be rapidly accelerated by the northward electric field, and the heavier ions would be regarded as essentially remaining northward or unchanged.**

To calculate the chemical reaction rate, we assumed a pre-existing concentration of 100 cm$^{-3}$ for $[Na^+ \cdot CO_2]$, and used the observed value of 3.1 MHz for foE$_S$ (which means $N_e = 1.2 \times 10^5$ cm$^{-3}$), and 170 K for T. The calculated rate is $v = 13$ cm$^{-3}$s$^{-1}$, in accord with the required source strength of sodium atoms of 3 sodium atoms cm$^{-3}$s$^{-1}$ for the formation of Na$_S$ (Cox et al., 1993). If we deduct some influences by eddy diffusion and loss of sodium atoms, this chemical rate can generate one Na$_S$ within several minutes. So no matter how many Na$^+$ ions are contained in E$_S$, the electrons in E$_S$ are always sufficient to produce Na$_S$. Perhaps that is why we often observed that even a very weak E$_S$ is always accompanied by Na$_S$ (Dou et. al., 2010).

We have added a new section 3.2 to discuss the E$_S$ and Na$_S$ described in this reply in details for our revised manuscript.

Other matters to address:

1. The Na lidar measurements in Figure 1(a) and 1(b) have concentrations varying by a factor of 2, even though co-located in Hefei. This cannot be correct. Even if the lidars have different vertical resolutions, the integrated Na density across the Nas layer should be the same.

Thanks for another valuable comment. We first apologize for the confusions and imprecisions in the data analysis. We have now thoroughly checked the raw data files of the wideband fluorescence resonance lidar and the temperature/wind lidar. We have learnt more about the data reduction/inversion methods for both lidars. The key conflict could possibly derived from the wideband lidar. Fig. 2* (a) shows the lidar system operated poorly after the midnight. But the first author made a reckless mistake in handling the bad data files (i.e., she accidentally deleted the bad file). Yet we can note that the signal to noise ratio (SNR) in the first half of the night is 10, still larger than the limitation of 2.

[Figure]

**Fig. 2\* (a) The entire sodium density profile observed by the wideband lidar on June 3rd, 2013. (b) The sodium density profile by T/W lidar.**

For the wideband sodium fluorescence resonance lidar, the inversion formula for the sodium number density $N$ at an altitude of $z$ is given as follows:

$$N = \frac{\sigma_R n_a(z_0)}{\sigma_{N_a}} \cdot \frac{(P(z)-P_B)z^2}{(P(z_0)-P_B)z_0^2} , \qquad (7)^* \qquad \text{(Xue, 2007)}$$

where $\sigma_R$ is the Rayleigh backscatter cross section,

$n_a(z_0)$ is the atmosphere density at a reference altitude, given by atmospheric model,

$\sigma_{N_a}$ is the effective sodium backscatter cross section,

$P(z)$ is the number of photons detected in the range interval (z-Δz/2, z+Δz/2),

$P_B$ is the expected photon count per range bin due to background signal and dark counts, calculated through the averaged background signal above 130 km, and,

$P(z_0)$ is the Rayleigh photon count at 30 km altitude, estimated by averaging the measured photon count over a 5-km range interval centered at 30 km (Gardner et al., 1986).

Among all the parameters, the variables are $\sigma_R$, $P(z)$, $P_B$, $P(z_0)$, and $z$ (means $N = f(\sigma_R, P(z), P_B, P(z_0), z)$. The error transfer formula of $N$ equals to:

$$\frac{\Delta N}{N} = \left|\frac{\partial \ln f}{\partial \sigma_R}\right| \Delta \sigma_R + \left|\frac{\partial \ln f}{\partial P(z)}\right| \Delta P(z) + \left|\frac{\partial \ln f}{\partial P_B}\right| \Delta P_B + \left|\frac{\partial \ln f}{\partial P(z_0)}\right| \Delta P(z_0) + \left|\frac{\partial f}{\partial z}\right| \Delta z.$$

(8)*

The fourth term, $\left|\frac{\partial \ln f}{\partial P(z_0)}\right| \Delta P(z_0) = \left|\frac{1}{P(z_0)-P_B}\right| \Delta P(z_0)$, makes the final error inversely proportional to the absolute value of $P(z_0) - P_B$. The averaged photon count $P(z_0)$ at 30 km is given by:

$$P(z_0) = \eta T_A^2 \frac{\lambda J}{hc} \frac{A_R}{4\pi z_0^2} \sigma_R \Delta z n_a(z_0),$$

(9)*

where $\eta$ is the overall system efficiency and,

$T_A$     one-way atmospheric transmittance of the lower atmosphere;

$\lambda$     optical wavelength, $0.589 \times 10^{-6}$ m;

$J$     laser pulse energy, J;

$h$     Planck's constant, $6.63 \times 10^{-34}$ J s;

$c$     velocity of light, $3 \times 108$ m/s;

$A_R$     receiver aperture area, m$^2$;

$n_a(z_0)$     the atmosphere density at $z_0$ (Gardner et al., 1986).

To minimize shot noise, $P(z_0)$ at 30 km altitude is estimated by averaging the measured photon count over a 5-km range interval centered at 30 km (Gardner et al., 1986). Because the atmospheric density decreases approximately exponentially with altitude, the average photon count at 30 km is computed by first subtracting the estimated background count, multiplying the result by $z^2$, taking the natural logarithm and then averaging over the range 27.5 km to 32.5 km (Gardner et al., 1986). We can

see $P(z_0)$ is therefore sensitively influenced by the background atmosphere and lidar system conditions.

[Figure]

**Fig. 3\* (a) Sodium density profile detected by the wideband lidar on May 5th, 2013. (b) Sodium density profile detected by the T/W lidar on that day.**

In comparison, we show that the sodium density profile detected on May 5th, 2013, by the wideband lidar is close to that by the T/W lidar (Fig. 3\* (a) and (b)). Both profiles exhibit a peak density of about 4000 cm$^{-3}$.

Then we check the raw data files on June 3rd (the case we have chosen in the manuscript) and May 5th, 2013 (for comparison). Through $h = c \times t/2$, the reference height of $z_0 = 30km$ is equivalent to $t \approx 0.2ms$ (marked by the light line at .2000ms). On June 3rd, $P(z_0) = 106$ (highlighted by the red circle in Fig. 4\* (a)), and the expected photocount at 130 km equals to 18 (not shown in Fig. 4\*(a), but could be read from the data file). On May 5th, $P(z_0) = 501$, and $P_B = 7$ (Fig. 4\*(b)). Since the error term is inversely proportional to the absolute value of $P(z_0) - P_B$, a much smaller $|P(z_0) - P_B|$ (about 5.6 times less) would cause the deduced sodium number density $N$ to increase.

[Figure]

**Fig. 4\* (a) The raw data file at 14:28 UT on June 3$^{rd}$. $P(z_0) = 106$ (highlighted by the red circle), and the expected photon count at 130 km equals to 18. (b) The raw data file at 17:41 UT on May 5$^{th}$, 2013. $P(z_0) = 501$, and $P_B = 7$.**

On the other hand, for the narrowband T/W lidar, the number of photons received by telescope from the range (z-Δz/2, z+Δz/2) is given by:

$$N(z, \nu_L, T, V) = \left(\frac{E_L}{hc/\lambda}\right) \times (\eta T_A^2) \times \left(\rho_{N_A}(z)\sigma_{SB}(\nu_L, T, V)\Delta z\right) \times \left(\frac{A_R}{z^2}\right) \times (T_\uparrow T_\downarrow) + N_B,$$

where                                                                                    (10)\* (Li, 2005)

  $\nu_L$          Transmitter laser frequency;

  $T$              Temperature (K);

  $V$              Wind velocity (m/s);

$E_L$            Transmitted laser pulse energy (J);

$\eta$            System efficiency;

$\rho_{N_A}(z)$      Na number density (m$^{-3}$);

$T_\uparrow$            Upward transmission in the Na layer;

$T_\downarrow$            Downward transmission in the Na layer.

Different from the wideband lidar, the deduced sodium number density $\rho_{N_A}(z)$ here is independent on the atmospheric conditions at a relative altitude $z_0$. Perhaps that is the reason why they establish a necessary T/W lidar nearby.

       We have added more detailed explanation on the discrepancy of the peak densities, around lines 5 to 18 on page 4. We have redrawn Fig. 1(a) on page 20 of the revised manuscript.

2. The statement on page 6 (line 28): "Thus, we conclude that the ionospheric echoes and the lightning activities exhibit an obvious synchronous behaviour." Presumably Es layers are observed over Wuhan in the absence of lightning activity. So the implication of this statement is that whenever strong lightning is present Es is observed – is that correct?

Thanks for the comment. From a lot of references, we can propose that lightnings causing influences on the ionosphere. However, we cannot make a definitive claim that lightnings would cause E$_S$ all the time, based on our current case study. More statistical work is needed for the further studies. Therefore, we have modified the absolute descriptions around lines 14 to 15 on page 7.

Minor issues:

page 2, line 4: I don't think the MLT is the "least known part of our planet" – what about the deep oceans? I think you mean "our planet's atmosphere".

Thanks for the comment. We have modified it to "our planet's atmosphere", and added a reference accordingly, around lines 3 to 4 on page 2. This part is indeed worth our lifetime of learning and relearning.

page 2, line 6: changed "sodium species" to "sodium atoms", since that is the form of Na that can be observed from the ground.

Thanks for the comment. We have changed "sodium species" to "sodium atoms", around line 7 on page 2.

page 2, lines 15-30: in this discussion there is no mention of the magnetic field, which is part of the classical $V \times B$ mechanism for sporadic E formation

Thanks for the comment. We have added the influences on $E_S$ by geomagnetic field around lines 23 to 24 on page 2.

page 4, line 10: "prefers the Es mechanism" implies that the Nas is an intelligent being that can make a choice! I would rephrase "is better explained by the Es mechanism"
Thanks for the comment. We have modified it to "this $Na_S$ is better explained by the $E_S$ mechanism" around lines 28 to 29 on page 4.

page 4, line 16: "suggests"
Thanks for the comment. We have modified it to "suggests" around line 1 on page 5 in the revised manuscript.

page 5, line 4: provide a citation for the equation
Thanks for the comment. We have added the citation of this equation around line 24 on page 5 in the revised manuscript. On page 10 of Bittencourt (2004), the electron plasma frequency is given by

$$\omega_{pe} = \left(\frac{n_e e^2}{m_e \epsilon_0}\right)^{1/2}, \qquad\qquad (11)*$$

where $n_e$

$m_e$
$\epsilon_0$

When $\omega_{pe}$ equals to the critical frequency detected by the ionosonde (foE$_S$), $n_e = \frac{m_e \epsilon_0}{e^2} \omega_{pe}^2 = \frac{m_e \epsilon_0}{e^2} foE_S^2 = 1.24 \times 10^4 foE_S^2 (cm^{-3})$.

page 5,line 29: the phrase "could be supported by a classic electrodynamics textbook" should be omitted. Either this is very well understood by the community, or you should provide reference to such a textbook.
Thanks for the comment. We have added citation of the textbook around lines 17 to 18 on page 6.

page 6, line 6: I cannot believe that the energy of this lightning stroke is known to 7 significant figures!
Thanks for the comment. We are so sorry for the misunderstanding of unit. We have checked the data description for WWLLN, and the unit of the power of one lightning stroke must be kW (reshown below as a screenshot). So these lightning strokes might be powerful enough to make contributions to ionospheric disturbances. We have modified the unit for lightning energy to be in kW throughout the manuscript.

```
Format of data files:
    Files are ascii text files labeled with the date and time of the data such as A20100203.loc for
data from 2010/02/03.    All data in UTC sample:
    2012/4/4,00:00:01.369657, 14.9924, -091.3758, 25.1, 8, 1109.25, 295.14, 6
    2012/4/4,00:00:01.479687, 14.7209, -087.7137, 10.1, 5, 1243.98, 431.82, 5
    2012/4/4,00:00:01.444158, 06.0155, 097.2494, 03.9, 5, 1004.87, 249.15, 3
    2012/4/4,00:00:01.534491, -17.6947, 040.8280, 16.2, 6, 5534.23, 4172.98, 3
    ...
    Year/Mo/Da,Hr:Mn:Sec.fract, Lat , Long, Resid, Nsta, power (kW), power uncertainty (kW), nstn
power
```

page 6, line 12: the shading in the figure is too faint to see.

Thanks for the comment. We have redrawn Figure 4. We hope the current figure
would be better.

page 6, line 17: the statement "such an idea/picture has been proposed long time ago"
must be referenced.

Thanks for the comment. We have added the references around line 4 on page 7.

page 6, line 30: the statement "The Sodium ions and electrons recombine much faster
during the overturning period, leading to a depletion of the Es" needs further
discussion. As I state above, I think this can only happen if the Es descends rapidly.
Another possibility is that somehow the Es becomes compressed so that the plasma
concentration increases – that seems unlikely.

Thanks for the comment. We wholeheartedly agree with this accelerated descending
mechanism. We have added a new section 3.2 to discuss the mechanism in the revised
manuscript.

page 7, line 12: change "consequential occurrence" to something like "consequent
production"

Thanks for the comment. We have changed it around line 22 on page 9.

page 9, line 5: the journal requires that the data is archived and accessible to the
reader. Not through writing to someone.

Thanks for the comment. We have changed it to "National Space Science Data
Center, National Science & Technology Infrastructure of China
(http://www.nssdc.ac.cn)".

page 19. Figure Caption 3: change "another Nas companying" to "another Nas being
produced" Figure 4 should be redrawn. The site of the electric mill is hidden. The
units of power are W, not J, and these "powers" contain too many significant figures.
The map and shading are too faint to read.

Thanks for the comment. We have changed the unit for power to kW. We have

redrawn Figure 4.

Figure 5. The legend in the figure needs to be explained in the figure caption.
Thanks for the comment. We have added the description in the figure caption of Fig. (5).

**Cited References for this Reply:**

Collins, S. C., Plane, J. M. C., Kelleya, M. C., Wright, T. G., Soldán, P., Kanee, T. J., Gerrarde, A. J., Grime, B. W., Rollason, R. J., and Friedman, J. S.: A study of the role of ion-molecule chemistry in the formation of sporadic sodium layers, Journal of Atmospheric & Solar Terrestrial Physics, 64, 845-860, 2002.

Cox, R. M., and Plane, J. M. C.: An ion-molecule mechanism for the formation of neutral sporadic Na layers, Journal of Geophysical Research Atmospheres, 103, 6349-6359, 1998.

Gardner, C. S., Voelz, D. G., Sechrist, C. F., and Segal, A. C.: Lidar studies of the nighttime sodium layer over Urbana, Illinois: 1. Seasonal and nocturnal variations, Journal of Geophysical Research Space Physics, 91, 13659-13673, 1986.

Dou, X. K., Xue, X. H., Li, T., Chen, T. D., Chen, C., and Qiu, S. C.: Possible relations between meteors, enhanced electron density layers, and sporadic sodium layers, Journal of Geophysical Research: Space Physics, 115, A06311, 2010.

Griffiths, D.J..: Introduction to Electrodynamics, 3rd ed., Prentice- Hall, Upper Saddle River, New Jersey, 1999.

Li,T.: SODIUM LIDAR OBSERVED VARIABILITY IN MESOPAUSE REGION TEMPERATURE AND HORIZONTAL WIND: PLANETARY WAVE INFLUENCE AND TIDAL-GRAVITY WAVE INTERACTIONS, Ph.D. thesis, Colorado State University, Fort Collins, Colorado, 48-50pp, 2005.

Ma, J., Xue, X., Dou, X., Chen, T., Tang, Y., Jia, M., Zou, Z., Li, T., Fang, X., and Cheng, X.: Large-Scale Horizontally Enhanced Sodium Layers Coobserved in the Midlatitude Region of China, Journal of Geophysical Research: Space Physics, 124, A026448, 2019.

Shukla, P. K. and Mamun, A. A.: Introduction to Dusty Plasma Physics, Institute of Physics Publishing, 2002.

Williams, B. P., Berkey, F. T., Sherman, J., and She, C. Y.: Coincident extremely large sporadic sodium and sporadic E layers observed in the lower thermosphere over Colorado and Utah, Annales Geophysicae, 25, 3-8, 2007.

Xue, X.H.: Studies on Geoeffectiveness of Coronal Mass Ejections and Near-Earth

Space Environment, Ph.D. thesis, University of Science & Technology of China, Hefei, China, 85-92pp, 2007.

---

## Author Comment (AC2) · 15 Mar 2021

**General comments:**

A reasonable mechanism of Nas layer enhancement in MLT region by lower atmospheric electric field was proposed in this manuscript, based on the observation date of several kinds of detection tools. This paper also provided a detail process how lower atmospheric electric field influences the ionized and neutral components in the upper atmosphere. This research is so novel and the similar reports are quite little up towards now. Consequently this manuscript fitted the scope of ACP at the moment in my point of view, however the mechanism of Nas layer enhancement in this manuscript needs to be further analyzed in the proceeding steps.

We would like to thank the reviewer for the valuable comments and constructive suggestions. We have studied all comments carefully and revised the manuscript accordingly. We marked all the changes in red fonts in the revised manuscript. The point-by-point answers to the comments are given below in blue fonts.

**Specific comments:**

1. In Fig.1a, the peak density of Nas is more than 12000 cm$^{-3}$ on 97.75 km at 14:40 UT (line 142). But it can be seen from Fig.1b, the peak density of Nas at the same time is only about 5000 cm$^{-3}$. Although the sodium density in Fig.1a and Fig.1b were given by two different kinds of lidars, the large difference of the density is difficult to be understood as these two lidars almost located at the same site. Is this large difference caused by the different resolutions or reversion methods? Authors should re-calculate the density in Fig.1a and Fig.1b, and give reasonable values.

Thanks for this valuable comment. We apologize on the confusions and imprecisions in the data analysis. We have now thoroughly checked the raw data files of the wideband fluorescence resonance lidar and the temperature/wind lidar. We have learnt more about the data reduction/inversion methods for both lidars. The key conflict could possibly come from the wideband lidar. Fig. 1*(a) shows the lidar system operated poorly after the midnight. But the first author made a reckless mistake in handling the bad data files (i.e., she accidentally deleted the bad file). Yet we can note that the signal to noise ratio (SNR) in the first half of the night is 10, still

larger than the limitation of 2.

[Figure]

**Fig. 1\* (a) The entire sodium density profile observed by the wideband lidar on June 3rd, 2013. (b) The sodium density profile by T/W lidar.**

For the wideband sodium fluorescence resonance lidar, the inversion formula for the sodium number density $N$ at an altitude of $z$ is given as follows:

$$N = \frac{\sigma_R n_a(z_0)}{\sigma_{Na}} \cdot \frac{(P(z)-P_B)z^2}{(P(z_0)-P_B)z_0^2} \; , \qquad (1)\* \qquad \text{(Xue, 2007)}$$

where $\sigma_R$ is the Rayleigh backscatter cross section,

$n_a(z_0)$ is the atmosphere density at a reference altitude, given by atmospheric model,

$\sigma_{Na}$ is the effective sodium backscatter cross section,

$P(z)$ is the number of photons detected in the range interval (z-$\Delta z$/2, z+$\Delta z$/2),

$P_B$ is the expected photon count per range bin due to background signal and dark counts, calculated through the averaged background signal above 130 km, and,

$P(z_0)$ is the Rayleigh photocount at 30 km altitude, estimated by averaging the measured photon count over a 5-km range interval centered at 30 km (Gardner et al., 1986).

Among all the parameters, the variables are $\sigma_R$, $P(z)$, $P_B$, $P(z_0)$, and $z$ (means $N = f(\sigma_R, P(z), P_B, P(z_0), z)$. The error transfer formula of $N$ equals to:

$$\frac{\Delta N}{N} = \left|\frac{\partial \ln f}{\partial \sigma_R}\right| \Delta \sigma_R + \left|\frac{\partial \ln f}{\partial P(z)}\right| \Delta P(z) + \left|\frac{\partial \ln f}{\partial P_B}\right| \Delta P_B + \left|\frac{\partial \ln f}{\partial P(z_0)}\right| \Delta P(z_0) + \left|\frac{\partial f}{\partial z}\right| \Delta z.$$

$$(2)\*$$

The fourth term, $\left|\frac{\partial \ln f}{\partial P(z_0)}\right| \Delta P(z_0) = \left|\frac{1}{P(z_0)-P_B}\right| \Delta P(z_0)$, makes the final error inversely proportional to the absolute value of $P(z_0) - P_B$. The averaged photon count $P(z_0)$ at 30 km is given by:

$$P(z_0) = \eta T_A^2 \frac{\lambda J}{hc} \frac{A_R}{4\pi z_0^2} \sigma_R \Delta z n_a(z_0), \qquad (3)*$$

where $\eta$ is the overall system efficiency and,

$T_A$      one-way atmospheric transmittance of the lower atmosphere;

$\lambda$      optical wavelength, $0.589 \times 10^{-6}$ m;

$J$      laser pulse energy, J;

$h$      Planck's constant, $6.63 \times 10^{-34}$ J s;

$c$      velocity of light, $3 \times 108$ m/s;

$A_R$      receiver aperture area, m²;

$n_a(z_0)$    the atmosphere density at $z_0$ (Gardner et al., 1986).

To minimize shot noise, $P(z_0)$ at 30 km altitude is estimated by averaging the measured photon count over a 5-km range interval centered at 30 km (Gardner et al., 1986). Because the atmospheric density decreases approximately exponentially with altitude, the average photon count at 30 km is computed by first subtracting the estimated background count, multiplying the result by $z^2$, taking the natural logarithm and then averaging over the range 27.5 km to 32.5 km (Gardner et al., 1986). We can see that $P(z_0)$ is therefore sensitively influenced by the background atmosphere and lidar system conditions.

[Figure]

**Fig. 2\* (a) Sodium density profile detected by the wideband lidar on May 5th, 2013. (b) Sodium density profile detected by the T/W lidar on that day.**

In comparison, we show that the sodium density profile detected on May 5th, 2013, by the wideband lidar is close to that by the T/W lidar (Fig. 2\* (a) and (b)). Both profiles exhibit a peak density of about 4000 cm$^{-3}$.

Then we check the raw data files on June 3rd (the case we have chosen in the manuscript) and May 5th, 2013 (for comparison). Through $h = c \times t/2$, the reference height of $z_0 = 30km$ is equivalent to $t \approx 0.2ms$ (marked by the light line at .2000ms). On June 3rd, $P(z_0) = 106$ (highlighted by the red circle in Fig. 3\* (a)), and the expected photocount at 130 km equals to 18 (not shown in Fig. 3\*(a), but could be read from the data file). On May 5th, $P(z_0) = 501$, and $P_B = 7$ (Fig. 3\*(b)). Since the error term is inversely proportional to the absolute value of $N(z_0) - N_B$, a much smaller $|P(z_0) - P_B|$ (about 5.6 times less) would cause the deduced sodium number density $N$ to increase.

On the other hand, for the narrowband T/W lidar, the number of photons received by telescope from the range (z-Δz/2, z+Δz/2) is given by:

$$N(z, \nu_L, T, V) = \left(\frac{E_L}{hc/\lambda}\right) \times (\eta T_A^2) \times \left(\rho_{N_A}(z)\sigma_{SB}(\nu_L, T, V)\Delta z\right) \times \left(\frac{A_R}{z^2}\right) \times (T_\uparrow T_\downarrow) + N_B,$$

where                                                  (4)\* (Li, 2005)

|  |  |
|---|---|
| $\nu_L$ | Transmitter laser frequency; |
| $T$ | Temperature (K); |
| $V$ | Wind velocity (m/s); |
| $E_L$ | Transmitted laser pulse energy (J); |
| $\eta$ | System efficiency; |
| $\rho_{N_A}(z)$ | Na number density (m$^{-3}$); |
| $T_\uparrow$ | Upward transmission in the Na layer; |
| $T_\downarrow$ | Downward transmission in the Na layer. |

Different from the wideband lidar, the deduced sodium number density $\rho_{N_A}(z)$ here is independent on the atmospheric conditions at a relative altitude $z_0$. Perhaps that is why they establish a necessary T/W lidar nearby.

[Figure]

**Fig. 3\* (a) The raw data file at 14:28 UT on June 3rd.** $P(z_0) = 106$ **(pointed out by the red circle), and the expected photocount at 130 km equals to 18. (b) The raw data file at 17:41 UT on May 5th, 2013.** $P(z_0) = 501$**, and** $P_B = 7$**.**

We have added more detailed explanation on the discrepancy of the peak densities, around lines 5 to 18 on page 4. We have redrawn Fig. 1(a) on page 20 of the revised manuscript.

2. In the Discussions, a clear causal chain is given by the authors: the lightning strokes induced the overturning of the electric field, and then induced the ionospheric disturbances, as well as Nas. However, it can be seen from Fig.2a: there is an

enhancement in the Es layer from 13:20 to 14:20, and the origin of this enhancement was not explained or discussed in the manuscript. Is it caused by lightning as proposed by Johnson and Davis (GRL, 2006) ? As the main contribution of this manuscript is to propose a new mechanism of Nas layer enhancement by lower atmospheric electric field, I suggest that Authors could explain or discuss the enhancement in the Es layer from 13:20 to 14:20.

Thanks for the comment. We would like to explain the $E_S$ enhancement through electron acceleration by the reverse electric field. Usually, the mid-latitude $E_S$ layers would be brought down gradually by tidal fluctuations (Mathews, 1998). Professor Plane's $E_S$ theory predicts that when a series of $E_S$ layers descend below 100 km, they will be depleted through recombination of ions and electrons (Cox and Plane, 1998). Since the recombination of $Na^+ + e^- \rightarrow Na + h\nu$ is inefficient in generating Nas, $Na^+$ is believed to first form a ligand $Na^+ \cdot N_2$ through the recombination reaction:

$$Na^+ + N_2 + M \rightarrow Na^+ \cdot N_2 + M, \qquad (5)*$$

with a rate coefficient of $k_1 = 4.8 \times 10^{-30}(T/200)^{-2.2}$ cm$^6$ molecule$^{-2}$ s$^{-1}$ (Cox and Plane, 1998). $Na^+ \cdot N_2$ can either switch with CO$_2$ (which will undergo dissociative electron recombination to form $Na$), or O (which reforms $Na^+$)(Cox and Plane, 1998). So the key factor of $E_S$ mechanism depends on the ratio of [O]/ [CO$_2$]. Recombination of $Na^+ \cdot CO_2$ and $e^-$ will increase rapidly as [O]/ [CO$_2$] decreases below the value of 100 (Cox and Plane, 1998). Then the sodium atoms could be formed directly from the following chemical reaction:

$$Na^+ \cdot CO_2 + e^- \rightarrow Na + CO_2. \qquad (6)* \text{ (Cox and Plane, 1998)}$$

The chemical reaction rate ($v$) for this second-order reaction could be calculated using the following equation:

$$v = k[Na^+ \cdot CO_2]N_e, \qquad (7)*$$

The reaction rate coefficient $k_2$ for the chemical reaction is experimentally measured to be:

$$k_2 = 1 \times 10^{-6}\sqrt{\frac{200}{T}} \quad (\text{cm}^6 \text{ molecule}^{-2} \text{ s}^{-1}) \qquad (8)*$$

(Collins et al., 2002; Cox and Plane, 1998; Daire et al., 2002), and the electron density

$N_e$ can be calculated using the following equation:

$$N_e = 1.24 \times 10^4 foE_S^2 \ (cm^{-3}) \qquad (9)^* \ (Bittencourt, 2004).$$

Overall, this $E_S$ mechanism is most widely accepted, if we neglect $k_1$ as being too small with an order of $10^{-30}$. A possible adaptation is to assume a plenty quantity of pre-existing $Na^+ \cdot N_2$ / $Na^+ \cdot CO_2$ in the sodium layer, and the $E_S$ just needs to provide enough additional electrons. Figure 4* (d) shows $E_S$ descending near 100 km at about 13:20 (marked by the blue dashed line). Then the $E_S$ depletes, and a moderate enhancement of Na occurs from 13:30 UT to 14:00 UT (pointed out by the green arrow in Figure 4*(a)). This sodium increase exhibits no obvious peak, which could probably be in accord with a normally descending $E_S$ governed by tides. In comparison, the peak profile of the $Na_S$ shows intense enhancement and sharp peak, indicating a distinct mechanism.

[Figure]

**Figure 4* (a) Time series of sodium density variations. (b) Atmospheric electric field**

variations. **(c) foE$_S$ variations from 13:00UT to 16:00 UT. (d) Altitudes of E$_S$. (Blue and red dashed lines and green arrow are discussed in this reply, see the text).**

Furthermore, a link between the electric field reverse and E$_S$ enhancement at 14:40 UT (marked by the red dashed line) could be established through the acceleration of electrons. Normally, positive particles will move along the direction of electric field, and negative particles do the opposite (Griffiths, 1999). Since metal ions are much heavier than electrons, the ions would drag electrons in order to move/drift together, this process is called the bipolar diffusion (Griffiths, 1999). In the initial stage, ions and electrons descend gradually under the southward electric field. In a partially ionized plasma, the characteristic frequencies for ions and electrons are associated with the collisions of the plasma particles with stationary neutrals (e.g., the electron–neutral collision frequency $v_{en}$ and the ion–neutral collision frequency $v_{in}$). The collision frequency $v_{sn}$ for scattering of the plasma species $s$ by the neutrals is

$$v_{sn} = n_n \sigma_s^n V_{Ts}, \qquad (10)\text{* (Shukla and Mamun, 2002)}$$

where $n_n$ is the neutral number density,

$\sigma_s^n$ is the scattering cross section (which is typically of the order of $5 \times 10^{-15}\,\text{cm}^2$ and depends weakly on the temperature $Ts$),

and $V_{Ts} = (k_B T_s/m_s)^{1/2}$ is the thermal speed of the species $s$.

So the relaxation times $\tau = \frac{1}{v}$ for ions and electrons are different in a partially ionized plasma, and electrons would respond much faster than the heavier sodium ions do (since $m_i \gg m_e$). At the moment when the electric field reverses, electrons will be rapidly accelerated by the northward electric field, and ions would be regarded as essentially remaining northward or unchanged. So no matter how many Na$^+$ ions are in E$_S$, the electrons in the E$_S$ are always sufficient to produce Na$_S$. Perhaps that is why we often observed that even a very weak E$_S$ is always accompanied by Na$_S$ (Dou et. al., 2010).

We have added a new section 3.2 to discuss the E$_S$ and Na$_S$ described in this reply in details for our revised manuscript.

3. From the progress of Nas in Fig.1a:The density of Nas increases with height drops after the Nas height lower than 100 km. A maximum of Nas is present at 97km around 13:30, and later the density of Nas decreases. But another maximum of Nas is present again around 14:20 (almost the same height), and the authors think that this Nas is produced by the overturn of low-altitude electric field. According to Plane's theory, when the Es drops below 100 km, Na ions in the Es are rapidly neutralized to form Nas. As the height decreases, the rate of Na ion neutralization increases. And so there's Nas maximum at about 13:20 UT around 97 km. Later, the Nas gradually weakened due to excessive consumption of Na ions. But for the bigger Nas around 14:20, there needs to be an implicit condition if the ions in the Es contribute. That is, Na ions in the Es increased at 13:30. This may be due to the addition of surrounding sodium ions resulting in Es increase. So there may be a possible mechanism: A reversal of the electric field adds sodium ions nearby, and these ions enhance Es. And then the sodium ions in the Es are neutralized to form Na atoms, with Es weakened.

Thanks for the comment. We agree with this enhancement mechanism described by the referee, except that we would like to replace the enhanced ions with electrons. We have added the possible mechanism in detail under the new section 3.2.

**Technical corrections:**

1. From line 22 to line 26 in the abstract: rewrite this long sentence.

Thanks for the comment. We have rewritten the sentence around lines 24 to 26 on page 1.

2. From line 47 to line 49, "the metal layers (especially the sodium layer), which located between about 80~110 km, could possibly act as a window to detect the MLT parameters by means of fluorescence resonance lidars." please add the corresponding reference.

Thanks for the comment. We have added the references around line 6 on page 2.

3. And from line 49 to line 51 please also add the reference.

Thanks for the comment. We have added the references around line 8 on page 2.

4. At the end of Line 64, please consider the word "candidate" whether is proper or not.

Thanks for the comment. We have changed it to "proposed mechanisms".

5. The author should pay much attention to the tense of the manuscript,the past tense shall be used when giving background information in the Abstract and Introduction, when describing methods used, and when presenting and discussing results. There are indeed quite a lot of such serious problems throughout all the manuscript. If possible, please ask a native speaker for help.

Thanks for the comment. We have tried our best to use appropriate tenses depending on the actual time of occurrences.

6. Line 137, please consider about the abbreviation "T/W lidar" as it appeared there for the first time.

Thanks for the comment. We have added the full name and a reference for the T/W lidar around line 4 on page 4.

7. Please change the line 162 as "in accord with our previous reports WHICH shown that an Nas higher than 96 km tended to be…

Thanks for the comment. We have modified this sentence around lines 28 to 29 on page 4.

8. Line 182, Based on the ABOVE observations

Thanks for the comment. We have modified it around line 5 on page 9.

9. Line 215 also POINTED out by the vertical red dashed line.

Thanks for the comment. We have rewritten the figure caption around line 4 on page 22.

10.Line 234 could be changed to "It is worth mentioned that…"

Thanks for the comment. We have changed it to "It is worth mentioning that" around line 32 on page 5.

11. Line 254 could be changed to "The electric field could change through within two distinct ways as below:"

Thanks for the comment. We have change it to "as below" around line 13 on page 6.

12.Line 291-292: "mainly concentrating in two ranges about (35.8ºN, 118.1ºE) and (25.1ºN, 113.8ºE)." please rewrite this sentence.

Thanks for the comment. We have modified this sentence in the figure caption 4 on page 23.

13.Line 302 could be changed to "Afterwards, no strong stroke WAS detected again in the discussed area."

Thanks for the comment. We have modified this sentence around line 13 on page 7.

14.Line 311-313 the caption of Figure 5 (a)~(l): please unify the tense of verbs and pay attention to the English writing again as I remind.

Thanks for the comment. We have rewritten the caption of Fig. (5).

15.Lines 335-336, "similar to how moving cars will crash in a traffic accident in the car in front suddenly turns back or brakes" Please rewrite this sentence within much more scientific aspect.

Thanks for the comment. We have deleted this scenario in the manuscript.

16.Line 338, are you sure by three steps? Please check it.

Thanks for the comment. We have modified the legend to be four steps in total, around lines 4 to 9 on page 9.

**Cited References for this Reply**

Collins, S. C., Plane, J. M. C., Kelleya, M. C., Wright, T. G., Soldán, P., Kanee, T. J., Gerrarde, A. J., Grime, B. W., Rollason, R. J., and Friedman, J. S.: A study of the role of ion-molecule chemistry in the formation of sporadic sodium layers, Journal of Atmospheric & Solar Terrestrial Physics, 64, 845-860, 2002.

Cox, R. M., and Plane, J. M. C.: An ion-molecule mechanism for the formation of neutral sporadic Na layers, Journal of Geophysical Research Atmospheres, 103, 6349-6359, 1998.

Gardner, C. S., Voelz, D. G., Sechrist, C. F., and Segal, A. C.: Lidar studies of the nighttime sodium layer over Urbana, Illinois: 1. Seasonal and nocturnal variations, Journal of Geophysical Research Space Physics, 91, 13659-13673, 1986.

Dou, X. K., Xue, X. H., Li, T., Chen, T. D., Chen, C., and Qiu, S. C.: Possible relations between meteors, enhanced electron density layers, and sporadic sodium layers, Journal of Geophysical Research: Space Physics, 115, A06311, 2010.

Griffiths, D.J.: Introduction to Electrodynamics, 3rd ed., Prentice- Hall, Upper Saddle River, New Jersey, 1999.

Li,T.: SODIUM LIDAR OBSERVED VARIABILITY IN MESOPAUSE REGION TEMPERATURE AND HORIZONTAL WIND: PLANETARY WAVE INFLUENCE AND TIDAL-GRAVITY WAVE INTERACTIONS, Ph.D. thesis, Colorado State University, Fort Collins, Colorado, 48-50pp, 2005.

Ma, J., Xue, X., Dou, X., Chen, T., Tang, Y., Jia, M., Zou, Z., Li, T., Fang, X., and Cheng, X.: Large-Scale Horizontally Enhanced Sodium Layers Coobserved in the Midlatitude Region of China, Journal of Geophysical Research: Space Physics, 124, A026448, 2019.

Mathews, J. D.: Sporadic E: current views and recent progress, Journal of Atmospheric and Solar-Terrestrial Physics, 60, 413-435, 1998.

Shukla, P. K. and Mamun, A. A.: Introduction to Dusty Plasma Physics, Institute of Physics Publishing, 2002.

Williams, B. P., Berkey, F. T., Sherman, J., and She, C. Y.: Coincident extremely large sporadic sodium and sporadic E layers observed in the lower thermosphere over Colorado and Utah, Annales Geophysicae, 25, 3-8, 2007.

Xue, X.H.: Studies on Geoeffectiveness of Coronal Mass Ejections and Near-Earth Space Environment, Ph.D. thesis, University of Science & Technology of China, Hefei, China, 85-92pp, 2007.

---

## Referee Report (RR1)

This version has been much improved than the previous one, and the authors gave a more reasonable analysis of the Nas layer enhancement in MLT region generated by lower atmospheric electric field. However, there are still some issues that need to be improved, which are listed as below:

1. Figure 1: Although the author provided inverting method of sodium density from two different kinds of lidar data in detail, the differences of sodium density between Figure 1a and Figure 1b are still quite large. The density of Figure 1a is almost double sizes of that of Figure 1b. And it is impossible to have such a big different result for two lidars which are almost in the same area (maybe less than 1km). Even though the author believes that this was the result of lidar noise affecting the density, but from Figure 3 in the "response to review2" provided by author, the sodium layer peak signal 107 is more than ten times than the noise (7). Although the count rate of this lidar is relatively lower, the density error is equal to the reciprocal of the square root of the signal minus the noise, which is about 1/10. So it is impossible for the signal-to-noise ratio to produce a double error consequently. The author should carefully consider about it, whether the inverted parameters (for example, the scattering cross section) were wrong processed? In addition, in "response to review2", as shown in Figure 2b "east density". Did that mean the density in Figure 1b is not vertical? This really makes me puzzled, I recommend the author to provide the density of the wind lidar in the vertical direction. The sodium densities detected by the oblique direction laser and the vertical direction laser are likely to be much different, as the distance between these two lasers is 40-50km in the height of the sodium layer, which has already been found in other sodium wind lidar data.

2. In my last review comment, I wrote: "it can be seen from Fig.2a: there is an enhancement in the Es layer from 13:20 to 14:20, and the origin of this enhancement was not explained or discussed in the manuscript. Is it caused by lightning as proposed by Johnson and Davis (GRL, 2006) ? I suggest that Authors could explain or discuss the enhancement in the Es layer from 13:20 to 14:20."
In this revised manuscript, the author believes that: electrons will follow the northward electric field and accumulate, but the ions still move in the same direction due to the difference in collision frequency (At the moment when the electric field reverses, electrons will be rapidly accelerated by the northward electric field, and ions would be regarded as essentially remaining northward or unchanged). This explanation is a little bit vague. The author also said in the previous paragraph: "Since metal ions are much heavier than electrons, the ions would drag electrons in order to move/drift together". Then why the electrons in this place can break away from the bondage of the deionization and accumulation

with the movement of the ions? The point that reversal of electric field leads the enhancement of Es was not the crucial work of this article (The idea that the electric field reversal leads to the enhancement of Es has already been proposed by other authors, and the main contribution of this article is to discover and explain the generation of Nas). But I still hope that the author can provide a deeper explanation on this issue according to the previous research.

3. Referring to the generation of Nas at 14:20, the author believes that if the electron concentration in Es increased a lot, it can speed up the neutralization of sodium ions, and leading to the appearance of a new Nas peak. This explanation does really make sense. Since the reactants increase, the products must also increase. But I still stick to that there could be another possible contribution to the formation of Nas: the reversal of the electric field caused the nearby metal ions (including sodium ions) to join into Es, also resulting the increase of Es. And then the sodium ions in Es were neutralized to produce sodium atoms (at the same time Es was weakened). Anyway, though Plane's theory (Cox and Plane, 1998, JGR) indicates that metal ions have a very short lifetime below 100km, actually many calcium ions appeared below 100km and last several hours, which were already reported (Gerding et al., 2001, Annales Geophysicae; Raizada et al., 2012, JGR; Raizada et al., 2020 GRL). Therefore, the metal ions actually exist below 100km in my opinion.

4. Page 5 Line 24: The formula is wrong. Authors double check their calculating results here and elsewhere. The critical frequency $f_oEs$ should be given by

$$f_oEs = f_{pe} = \frac{\omega_{pe}}{2\pi} = \left(\frac{n_e e^2}{4\pi^2 m_e \varepsilon_0}\right)^{1/2}.$$

---

## Referee Report (RR2)

Manuscript title: Sporadic sodium layer: A possible tracer for the conjunction between the upper and lower atmospheres

Authors: Shican Qui, Ning Wang et al.

Manuscript no.: **acp-2020-1079**

The authors have dealt with many of the points raised in the initial round of reviews. However, there are still two major matters that need to be addressed further.

1. The mechanism for sporadic Na layer production

Both in the response to the reviewers, and in the revised manuscript, the authors have explored the ion-molecule mechanism for $Na_s$ production (Cox and Plane, 1989). They make a curious statement: "if we neglect $k_1$ as being too small with an order of $10^{-30}$ ". But if this reaction does not occur, how does $Na^+$ become neutralized?! I think the authors have not realised that this is a *third*-order reaction, so that the second-order rate coefficient is given by $k[N_2]$. Instead, they have assumed an arbitrary concentration of $Na^+.CO_2$ ions with a concentration of 100 $cm^{-3}$ – without any justification, other than the rate of production of Na atoms via dissociative recombination with electrons is of the right order to explain their observations.

I would suggest a major rewrite of this section of the paper. The following is reproduced from the supporting information to a recent paper (Jiao et al., *GRL*, 2017) from Plane's group in Leeds.

**Table S1. Ion-molecule reaction rate coefficients for Na** (adapted from *Plane et al.* [2015])

| No. | Reaction | Rate coefficient[a] |
|---|---|---|
| 1 | $Na + O_2^+ \rightarrow Na^+ + O_2$ | $2.7 \times 10^{-9}$ |
| 2 | $Na + NO^+ \rightarrow Na^+ + NO$ | $8 \times 10^{-10}$ |
| 3 | $Na^+ + N_2 (+ M = N_2 \& O_2) \rightarrow Na^+.N_2$ | $4.8 \times 10^{-30} (T/200 \text{ K})^{-2.2}$ |
| 4 | $Na^+ + CO_2 (+ M) \rightarrow Na^+.CO_2$ | $3.7 \times 10^{-29} (T/200 \text{ K})^{-2.84}$ |
| 5 | $Na^+.N_2 + CO_2 \rightarrow Na^+.CO_2 + N_2$ | $6 \times 10^{-10}$ |
| 6 | $Na^+.N_2 + O \rightarrow NaO^+ + N_2$ | $4 \times 10^{-10}$ |
| 7 | $NaO^+ + O \rightarrow Na^+ + O_2$ | $1 \times 10^{-11}$ |
| 8 | $NaO^+ + N_2 \rightarrow Na^+.N_2 + O$ | $1 \times 10^{-12}$ |
| 9 | $NaO^+ + O_2 \rightarrow Na^+ + O_3$ | $5 \times 10^{-12}$ |
| 10 | $NaO^+ + CO_2 \rightarrow Na^+.CO_2 + O$ | $6 \times 10^{-10}$ |
| 11 | $Na^+.X (X = O, N_2, CO_2) + e^- \rightarrow Na + X$ | $1 \times 10^{-6} (200 \text{ K}/T)^{0.5}$ |

[a] Units: bimolecular reactions, $cm^3$ molecule$^{-1}$ s$^{-1}$; termolecular reactions, $cm^6$ molecule$^{-2}$ s$^{-1}$

Application of reaction branching probabilities to reactions 3-11 yields the following first-order rate coefficient for the neutralization rate of $Na^+$ ions [*Plane*, 2004]:

$$k(Na^+ \rightarrow Na) = k_3[N_2][M] \times Pr(Na^+.N_2 \rightarrow Na) + k_4[CO_2][M]$$

$$k_3[N_2][M]\left(\frac{k_{11}[e^-]+k_5[CO_2]}{k_{11}[e^-]+k_5[CO_2]+k_6[O] \times Pr(NaO^+ \rightarrow Na^+)}\right) + k_4[CO_2][M]$$

$$= k_3[N_2][M]\left(\frac{k_{11}[e^-]+k_5[CO_2]}{k_{11}[e^-]+k_5[CO_2]+k_6[O]\left(\dfrac{k_7[O]+k_9[O_2]}{k_7[O]+k_8[N_2]+k_9[O_2]+k_{10}[CO_2]+k_{11}[e^-]}\right)}\right) + k_4[CO_2][M]$$

where Pr() denotes the branching probability.

Notes:

1. Formation of $Na^+.CO_2$ leads irreversibly to Na via DR, because this ion is thermally stable and does ligand-witch with O, $N_2$ or $O_2$ [*Cox and Plane*, 1998].

2. Once $NaO^+$ forms via reaction 6, reactions 7 and 9 convert it back to $Na^+$. However, these reactions are in competition with reactions 8, 10 and 11. Reactions 10 and 11 will always produce Na via DR; reaction 8 produces $Na^+.N_2$, so there is a small probability that this will recycle again through $NaO^+$ to $Na^+$ and not lead to DR.

3. Reactions involving $H_2O$ are not included (analogues to R4, R5, R10 and R11), because $[H_2O]$ is around 2 orders of magnitude lower than $[CO_2]$, and these species have very similar chemistries.

The expression for the first-order conversion rate of $Na^+$, $k(Na^+ \rightarrow Na)$, can easily be computed as a function of height in a spreadsheet using typical values for $N_2$, $O_2$, O etc., and the electron density from the ionosonde and temperature from the W-T lidar. Discussion of the vertical profile of $k(Na^+ \rightarrow Na)$ should replace the first part of Section 3.2, up to around line 10 on page 8. The $Na^+$ concentration required to produce the observed production rate of Na is simply obtained from d[Na]/dt = $k(Na^+ \rightarrow Na)[Na^+]$, and the question is whether this is sensible in terms of the expected fraction of $Na^+$ ions which make up the total of metallic ions in the $E_s$ layer.

**It is still not clear what the role of lightning in this process is**. The authors say that the field reversal induced by lightning will increase the electron density below 100 km by causing rapid downward transport of electrons. But does this actually happen? They do not quantify the magnitude of increase in electron density that might occur at 98 km, where the $Na_s$ layer is observed. A more important question is whether it is electrons that are rate-determining in the neutralization process. In the expression for $k(Na^+ \rightarrow Na)$, the question is whether $k_{11}[e^-]$ is larger than $k_5[CO_2]$. I do not believe that it is, so increasing electrons will make little difference to $k(Na^+ \rightarrow Na)$. An obvious piece of evidence is that $Na_s$ form without lightning being required.

2. The discrepancy between the two lidar records in Figure 1

Apparently there were problems with the USTC lidar shown in Figure 1. It is not clear to me what they were, and I still do not understand how the absolute Na density could vary by a factor of 2. The explanation on page 4 (lines 8 – 18) is unclear. It seems to be something to do with the uncertainty in the Na density retrieved from the broadband lidar. If the authors know what the problem is, they have two options: 1) discard the data, and just show data from the W/T lidar which makes up 3 of the 4 panels in Figure 1; 2) present a proper estimate of the uncertainty in the absolute density retrieved from each system, and show that they are in accord.

---

## Author Response (AR2)

We would like to thank the reviewers for the valuable comments and suggestions. We have studied all comments carefully and revised the manuscript accordingly. We mark all the final changes in red fonts in the revised manuscript. The point-by-point answers to the comments are given below in blue fonts.

**Responses to Reviewer # 1**

Manuscript title: Sporadic sodium layer: A possible tracer for the conjunction between the upper and lower atmospheres

Authors: Shican Qiu, Ning Wang et al.

Manuscript no.: **acp-2020-1079**

The authors have dealt with many of the points raised in the initial round of reviews. However, there are still two major matters that need to be addressed further.

1. The mechanism for sporadic Na layer production

Both in the response to the reviewers, and in the revised manuscript, the authors have explored the ion-molecule mechanism for Nas production (Cox and Plane, 1989). They make a curious statement: "if we neglect $k_1$ as being too small with an order of $10^{-30}$ ". But if this reaction does not occur, how does $Na^+$ become neutralized?! I think the authors have not realised that this is a *third*-order reaction, so that the second-order rate coefficient is given by $k[N_2]$. Instead, they have assumed an arbitrary concentration of $Na^+.CO_2$ ions with a concentration of 100 cm$^{-3}$ – without any justification, other than the rate of production of Na atoms via dissociative recombination with electrons is of the right order to explain their observations. I would suggest a major rewrite of this section of the paper. The following is reproduced from the supporting information to a recent paper (Jiao et al., *GRL*, 2017) from Plane's group in Leeds.

**Table S1. Ion-molecule reaction rate coefficients for Na** (adapted from *Plane et al.* [2015])

| NO. | Reaction | Rate Coefficient[a] |
|---|---|---|
| 1 | $Na + O_2^+ \rightarrow Na^+ + O_2$ | $2.7 \times 10^{-9}$ |

| 2 | $Na + NO^+ \rightarrow Na^+ + NO$ | $8.0 \times 10^{-10}$ |
|---|---|---|
| 3 | $Na^+ + N_2(+M = N_2 \ \& \ O_2) \rightarrow Na^+ \cdot N_2$ | $4.8 \times 10^{-30}(T \ / \ 200K)^{-2.2}$ |
| 4 | $Na^+ + CO_2(+M) \rightarrow Na^+ \cdot CO_2$ | $3.7 \times 10^{-29}(T \ / \ 200K)^{-2.84}$ |
| 5 | $Na^+ \cdot N_2 + CO_2 \rightarrow Na^+ \cdot CO_2 + N_2$ | $6.0 \times 10^{-10}$ |
| 6 | $Na^+ \cdot N_2 + O \rightarrow NaO^+ + N_2$ | $4.0 \times 10^{-10}$ |
| 7 | $NaO^+ + O \rightarrow Na^+ + O_2$ | $1.0 \times 10^{-11}$ |
| 8 | $NaO^+ + N_2 \rightarrow Na^+ \cdot N_2 + O$ | $1.0 \times 10^{-12}$ |
| 9 | $NaO^+ + O_2 \rightarrow Na^+ + O_3$ | $5.0 \times 10^{-12}$ |
| 10 | $NaO^+ + CO_2 \rightarrow Na^+ \cdot CO_2 + O$ | $6.0 \times 10^{-10}$ |
| 11 | $Na^+ \cdot X \ (X = O, N_2, CO_2) + e^- \rightarrow Na + X$ | $1.0 \times 10^{-6}(200K \ / \ T)^{0.5}$ |

a Units: bimolecular reactions, $cm^3$ molecule$^{-1}$ s$^{-1}$; termolecular reactions, $cm^6$ molecule$^{-2}$ s$^{-1}$

Application of reaction branching probabilities to reactions 3-11 yields the following first-order rate coefficient for the neutralization rate of $Na^+$ ions [*Plane*, 2004]:

$$k(Na^+ \rightarrow Na) = k_3[N_2][M] \times Pr(Na^+ \cdot N_2 \rightarrow Na) + k_4[CO_2][M]$$

$$= k_3[N_2][M] \left( \frac{k_{11}[e^-] + k_5[CO_2]}{k_{11}[e^-] + k_5[CO_2] + k_6[O] \times Pr(NaO^+ \rightarrow Na^+)} \right) + k_4[CO_2][M]$$

$$= k_3[N_2][M] \left( \frac{k_{11}[e^-] + k_5[CO_2]}{k_{11}[e^-] + k_5[CO_2] + k_6[O] \left( \frac{k_7[O] + k_9[O_2]}{k_6[O] + k_8[N_2] + k_9[O_2] + k_{10}[CO_2] + k_{11}[e^-]} \right)} \right)$$

$$+ k_4[CO_2][M]$$

where Pr() denotes the branching probability.

Notes:

1. Formation of $Na^+.CO_2$ leads irreversibly to Na via DR, because this ion is thermally stable and does ligand-witch with O, $N_2$ or $O_2$ [*Cox and Plane*, 1998].

2. Once $NaO^+$ forms via reaction 6, reactions 7 and 9 convert it back to $Na^+$. However, these reactions are in competition with reactions 8, 10 and 11. Reactions 10 and 11 will always produce Na via DR; reaction 8 produces $Na^+.N_2$, so there is a small probability that this will recycle again through $NaO^+$ to $Na^+$ and not lead to DR.

3. Reactions involving $H_2O$ are not included (analogues to R4, R5, R10 and R11), because $[H_2O]$ is around 2 orders of magnitude lower than $[CO_2]$, and these species have very similar chemistries. The expression for the first-order conversion rate of $Na^+$, $k(Na^+\rightarrow Na)$, can easily be computed as a function of height in a spreadsheet using typical values for $N_2$, $O_2$, $O$ etc., and the electron density from the ionosonde and temperature from the W-T lidar. Discussion of the vertical profile of $k(Na^+\rightarrow Na)$ should replace the first part of Section 3.2, up to around line 10 on page 8. The $Na^+$ concentration required to produce the observed production rate of Na is simply obtained from $d[Na]/dt = k(Na^+\rightarrow Na)[Na^+]$, and the question is whether this is sensible in terms of the expected fraction of $Na^+$ ions which make up the total of metallic ions in the $E$s layer.

**It is still not clear what the role of lightning in this process is**. The authors say that the field reversal induced by lightning will increase the electron density below 100 km by causing rapid downward transport of electrons. But does this actually happen? They do not quantify the magnitude of increase in electron density that might occur at 98 km, where the $Na_S$ layer is observed. A more important question is whether it is electrons that are rate-determining in the neutralization process. In the expression for $k(Na^+\rightarrow Na)$, the question is whether $k_{11}[e^-]$ is larger than $k_5[CO_2]$. I do not believe that it is, so increasing electrons will make little difference to $k(Na^+\rightarrow Na)$. An obvious piece of evidence is that $Na_S$ form without lightning being required.

Reply: We would like to note our appreciation for all the contributions that this kind and wise reviewer has made for our manuscript. We apologize for addressing the chemical reactions in such a cursory manner during the previous revision. The first-order rate coefficient for the neutralization rate of $Na^+$ ions, $k$ $(Na^+\rightarrow Na)$, has been calculated, with the help of WACCM-Na model simulation results from Dr. Wuhu Feng. We have modified section 3.2 substantially following the reviewer's comments.

On the other hand, we admit that we are only broadly sketching the lightnings as

a possible trigger of the electric field overturning. The electric field could turn upward through electrostatic induction by the thunderstorm. The strong lightnings could be regarded as an index of thunderstorm. And the echoes observed by the ionosonde exhibits synchronous activities with the lightnings, indicating a possible link between the lower atmosphere and the ionosphere. We hope to pursue some further study on the candidate triggers, such as lightnings, sprites, and elves. In any case, we would not doubt that the main source of $Na_S$ is $E_S$. These authors always support the $E_S$ mechanism most. The first author Shican Qiu and Xingjin Wang (her old student) are always dreaming of studying abroad sometime in Plane's group, if and when the concerns on coronal virus ends. For these flimsy reasons, we still hope to retain the contents about lightnings, without further significant new insights, in our manuscript.

[Figure]

Figure 1** Model simulation results from WACCM-Na. (a) Constituents of the species used for calculating $k$ ($Na^+\rightarrow Na$). The number densities of $CO_2$, $O_2$, O, the total atmosphere density $M$ and $N_2 \approx [M] - [O_2] - [O]$ are derived from Yuan et al., 2019. The number density of electrons equals to $e \approx 1.24 \times 10^4 f_o E_S^2$ $(cm^{-3})$. (b)

The calculated first-order rate coefficient $k$ (Na$^+$→Na), indicating much more efficient recombination below about 100 km.

2. The discrepancy between the two lidar records in Figure 1

Apparently there were problems with the USTC lidar shown in Figure 1. It is not clear to me what they were, and I still do not understand how the absolute Na density could vary by a factor of 2. The explanation on page 4 (lines 8 – 18) is unclear. It seems to be something to do with the uncertainty in the Na density retrieved from the broadband lidar. If the authors know what the problem is, they have two options: 1) discard the data, and just show data from the W/T lidar which makes up 3 of the 4 panels in Figure 1; 2) present a proper estimate of the uncertainty in the absolute density retrieved from each system, and show that they are in accord.

Reply: Thanks for this comment. We have checked the original data files carefully again, and found the sodium density by the west beam of T/W lidar would probably be about 8650 cm$^{-3}$. We have discarded the bad data by the wide band lidar and replaced with a new image for the west beam of T/W lidar. Thus, Fig. 3(a) has been modified accordingly.

[Figure]

Figure 2** Observations on June 3rd, 2013, by the USTC T/W lidar. (a) The sodium density profile of the west beam by T/W lidar A moderate increase of sodium density appears at about 13:20 UT, while the largest intensity of sodium enhancement begins at about 14:20 UT. The sodium density peaks at 14:37 UT around 97.65 km. (b) Temperature profile observed by the T/W lidar, showing a cold region where the $Na_S$ occurs. (c) The zonal wind detected by the T/W lidar, exhibiting a suitable wind shear for the creation or formation of $E_S$.

**Responses to Reviewer # 2**

This version has been much improved than the previous one, and the authors gave a more reasonable analysis of the Nas layer enhancement in MLT region generated by lower atmospheric electric field. However, there are still some issues that need to be improved, which are listed as below:

1. Figure 1: Although the author provided inverting method of sodium density from two different kinds of lidar data in detail, the differences of sodium density between Figure 1a and Figure 1b are still quite large. The density of Figure 1a is almost double sizes of that of Figure 1b. And it is impossible to have such a big different result for two lidars which are almost in the same area (maybe less than 1km). Even though the author believes that this was the result of lidar noise affecting the density, but from Figure 3 in the "response to review2" provided by author, the sodium layer peak signal 107 is more than ten times than the noise (7). Although the count rate of this lidar is relatively lower, the density error is equal to the reciprocal of the square root of the signal minus the noise, which is about 1/10. So it is impossible for the signal-to-noise ratio to produce a double error consequently. The author should carefully consider about it, whether the inverted parameters (for example, the scattering cross section) were wrong processed? In addition, in "response to review2", as shown in Figure 2b "east density". Did that mean the density in Figure 1b is not vertical? This really makes me puzzled, I recommend the author to provide the density of the wind lidar in the vertical direction. The sodium densities detected by the oblique direction laser and the vertical direction laser are likely to be much different, as the distance between these two lasers is 40-50km in the height of the sodium layer, which has already been found in other sodium wind lidar data.

Reply: We truly appreciate all the contributions this kind and wise reviewer has made in improving our manuscript. In fact, the wide band lidar provides the vertical sodium density, while the narrowband lidar observes the oblique direction. We have checked the original data files carefully again and found that the sodium density from the west beam of T/W lidar would probably be about 8650 $cm^{-3}$. So we see the deduced sodium

density are actually different for each lidar beam. Considering both reviewers' comments, we now think it is better to discard the bad data by the wide band lidar, since it operated poorly on that day. We have replaced with a new image with the results taken from the west beam of T/W lidar. Both Fig. 1(a) and Fig. 3(a) have been modified accordingly in the final revised manuscript.

[Figure]

Figure 1** Observations on June 3rd, 2013, by the USTC T/W lidar. (a) The sodium density profile of the west beam by T/W lidar A moderate increase of sodium density appears at about 13:20 UT, while the largest intensity of sodium enhancement begins at about 14:20 UT. The sodium density peaks at 14:37 UT around 97.65 km. (b) Temperature profile observed by the T/W lidar, showing a cold region where the NaS

occurs. (c)The zonal wind detected by the T/W lidar, exhibiting a suitable wind shear for the creation or formation of $E_S$.

2. In my last review comment, I wrote: "it can be seen from Fig.2a: there is an enhancement in the Es layer from 13:20 to 14:20, and the origin of this enhancement was not explained or discussed in the manuscript. Is it caused by lightning as proposed by Johnson and Davis (GRL, 2006) ? I suggest that Authors could explain or discuss the enhancement in the Es layer from 13:20 to 14:20." In this revised manuscript, the author believes that: electrons will follow the northward electric field and accumulate, but the ions still move in the same direction due to the difference in collision frequency (At the moment when the electric field reverses, electrons will be rapidly accelerated by the northward electric field, and ions would be regarded as essentially remaining northward or unchanged). This explanation is a little bit vague. The author also said in the previous paragraph: "Since metal ions are much heavier than electrons, the ions would drag electrons in order to move/drift together". Then why the electrons in this place can break away from the bondage of the deionization and accumulation with the movement of the ions? The point that reversal of electric field leads the enhancement of Es was not the crucial work of this article (The idea that the electric field reversal leads to the enhancement of Es has already been proposed by other authors, and the main contribution of this article is to discover and explain the generation of Nas). But I still hope that the author can provide a deeper explanation on this issue according to the previous research.

Reply: Thanks for this comment. The authors have made a primary statistical work, indicating the atmospheric electric field would probably influence the $E_S$ layers. However, the results exhibit complex features as the following images show. For some cases during the overturning of electric field, foE$_S$ may be detected to increase in the data. But during other cases the E$_S$ may disappear. Thus, we cannot draw a conclusion

on the definite pattern currently. The enhancement of $E_S$ during the lightnings was first proposed by Johnson and Davis (2006), without detailed explanation about the mechanism. And those authors do not give further research results on the enhancement of $E_S$. So, it is still beyond our capability and expertise to explain the mechanism of the enhancement of $E_S$ involved. We propose a further study on the link between electric field, lightnings, and $E_S$ in the future work.

[Figure]

Figure 2** A typical case of foE$_S$ increase and h'E$_S$ rise caused by electric field overturning was observed on June 22, 2012

[Figure]

Figure 3** A typical case of Eₛ disappearance caused by atmospheric electric field inversion was observed on May 21, 2012

On the other hand, under a quasi-equilibrium state, the ions and electrons would move together at least during the initial stage (e.g. the heavier ions dragging the electrons, which is called ambipolar diffusion). So the ions and electrons would move downward during a southward electric field. But in a nonequilibrium condition, e.g., at the point of the electric field overturning, each plasma species has a different relaxation time (the time needed for establishing equilibrium again). The relaxation for the electrons is much shorter than that for the ions, because $\tau = (4\pi\varepsilon_0)^2 \frac{3m^{1/2}T^{3/2}}{4\sqrt{2\pi}nq^4 \ln \Lambda}$ is positively related to the particle mass. This is why the electrons would respond much faster than the ions do during the phase of relaxation. This discrepancy would cause a charge separation temporarily. The electrons move opposite along the electric field,

which means during the upward electric field they would move downward. However, after the relaxation time, the system would reach an equilibrium state again. We have added some more explanations and descriptions for the relaxation time from line 7 to line 32 on page 8.

3. Referring to the generation of Nas at 14:20, the author believes that if the electron concentration in Es increased a lot, it can speed up the neutralization of sodium ions, and leading to the appearance of a new Nas peak. This explanation does really make sense. Since the reactants increase, the products must also increase. But I still stick to that there could be another possible contribution to the formation of Nas: the reversal of the electric field caused the nearby metal ions (including sodium ions) to join into Es, also resulting the increase of Es. And then the sodium ions in Es were neutralized to produce sodium atoms (at the same time Es was weakened). Anyway, though Plane's theory (Cox and Plane, 1998, JGR) indicates that metal ions have a very short lifetime below 100km, actually many calcium ions appeared below 100km and last several hours, which were already reported (Gerding et al., 2001, Annales Geophysicae; Raizada et al., 2012, JGR; Raizada et al., 2020 GRL). Therefore, the metal ions actually exist below 100km in my opinion.

Reply: Thanks for the comment. We have made a simulation of the chemical reactions through the encouragement of the other reviewer. The main reactions and corresponding rate coefficients for the sodium species under the mesopause condition are summarized in the new Table 2 of our revised manuscript (Cox and Plane, 1998; Jiao et al., 2017; Plane et al., 2015; Plane, 2004; Yuan et al., 2019). Application of reaction branching probabilities to reactions 3 to 11 yield the following first-order rate coefficient for the neutralization rate of $Na^+$ ions (Plane, 2004):

$$k(Na^+ \rightarrow Na) = k_3[N_2][M] \times Pr(Na^+ \cdot N_2 \rightarrow Na) + k_4[CO_2][M]$$

$$= k_3[N_2][M]\left(\frac{k_{11}[e^-]+k_5[CO_2]}{k_{11}[e^-]+k_5[CO_2]+k_6[O]\times Pr(NaO^+\rightarrow Na^+)}\right) + k_4[CO_2][M]$$

$$= k_3[N_2][M]\left(\frac{k_{11}[e^-]+k_5[CO_2]}{k_{11}[e^-]+k_5[CO_2]+k_6[O]\left(\frac{k_7[O]+k_9[O_2]}{k_6[O]+k_8[N_2]+k_9[O_2]+k_{10}[CO_2]+k_{11}[e^-]}\right)}\right)$$

$$+ k_4[CO_2][M]$$

where *Pr* denotes the branching probability. The first-order conversion rate of $k$ (Na$^+\rightarrow$Na) can be computed as a function of height using typical values for $N_2$, $O_2$, O, $CO_2$ from a WACCM-Na model simulation (Yuan et al., 2019). The calculated results are given by Figure 6. The simulation results show sodium ions could probably exist below 100 km: the inflection point of $k$ (Na$^+\rightarrow$Na) comes out around 100 km, and below that altitude, the sodium ions would recombine with electrons efficiently through cycling chemical under a large $k$ value. So the metal ions might actually exist below 100 km as the reviewer regards.

[Figure]

Figure 4** Model simulation results from WACCM-Na. (a) Constituents of the species used for calculating $k$ ($Na^+$→Na). The number densities of $CO_2$, $O_2$, O, the total atmosphere density $M$ and $N_2 \approx [M] - [O_2] - [O]$ are derived from Yuan et al., 2019. The number density of electrons equals to $e \approx 1.24 \times 10^4 f_o E_s^2 \ (cm^{-3})$. (b) The calculated first-order rate coefficient $k$ ($Na^+$→Na), indicating much more efficient recombination below about 100 km.

4. Page 5 Line 24: The formula is wrong. Authors double check their calculating results here and elsewhere. The critical frequency $foE_S$ should be given by

$$foE_S = f_{pe} = \frac{\omega_{pe}}{2\pi} = \left(\frac{n_e e^2}{4\pi^2 m_e \varepsilon_0}\right)^{\frac{1}{2}}$$

Reply: Thanks for the comment. We have modified this formula around line 9 on page 5.

---

## Author Response (AR3)

We would like to thank the reviewers for the valuable comments and suggestions. We have studied all comments carefully and revised the manuscript accordingly. We mark the major changes in red in the revised manuscript. The point-by-point answers to the comments are shown below in blue fonts.

**Referee #1**

This version of the paper is much improved because of the inclusion of the new part of Section 3, and Figure 6, which show that the observed $Na_S$ layer can probably be explained by the established ion-molecule chemistry of $Na^+$ ions in a descending sporadic E layer. Therefore, as the authors conclude, it is not necessary to invoke a mechanism involving lightning. However, this does not mean that lightning does not enhance $Na_S$ production. This is a novel idea and the publication of the paper will stimulate further interest - in particular, the regarding the precise way in which electric field reversal could cause or amplify $Na_S$ formation needs to be modelled properly, and this is beyond the scope of this paper.

I have several suggestions to improve the current version, but these are all minor revisions:

Abstract. These two sentences should be changed: "Our results suggest that lightning strokes would probably have an influence on the ionosphere and thus affecting the occurrence of $Na_S$, with the overturning of electric field playing an important role. Statistical results reveal that the sporadic E layers ($E_S$) could hardly be formed or maintained when the atmospheric electric field turns upward."

I suggest something like: "Our results suggest that a lightning stroke could trigger or amplify the formation of a $Na_S$ layer in a descending sporadic E layer, through a mechanism that involves overturning of the electric field. This is based on our observation that, statistically, sporadic E layers disappear rapidly when the atmospheric electric field turns upward".
Reply: Thanks for the comment. We have modified the two sentences in the abstract following the reviewer's suggestions.

page 7, line 4: "Normally", not "Nominally"
Reply: Thanks for the comment. We have modified this word.

page 7, line 21: "equates", not "equivalents"
Reply: Thanks for the comment. We have changed "equivalents" to "equates".

page 8, line 8: "do the opposite"
Reply: Thanks for the comment. We have added "the".

page 8, line 25: "The electrons would reverse rapidly before the ions can respond similar

to the velocity overshoot effect for electrons. During the relaxation phase, the 25 recombination between electrons and ions would probably be triggered through collisions, not unlike how moving cars will crash in a traffic accident if the car behind suddenly accelerates."

I do not understand the second sentence. Ion-electron recombination reactions, whether dissociative or radiative, have small negative temperature dependences i.e. the reactions get slower at higher temperatures, because the impact velocity increases and the long-range attraction between the ion and electron is less effective.
Reply: Thanks for the comment. We have modified the description according to the reviewer's suggestions.

Figure 6. This would be much clearer if you use a log scale for the abscissa (x-axis), for both (a) and (b)
Reply: Thanks for the comment. We have changed the x-axis to a logarithmic coordinate.

Thanks again for all the contributions made by this kind and wise reviewer.

**Referee #2**

I am very glad to see that the authors have answered almost all my questions and revised the manuscript under my suggestions. I also support that the authors added a simulation of the chemical reactions to explain the efficient neutralization of Na ions in this $Na_S$ case study, which is actually that the enough Na ions were neutralized to from $Na_S$ layer, as I have indicated in last referee report. Now the manuscript is rather perfect, but a few issues are needed to be considered:

1. Page 2 line 9, the authors wrote "With an active chemical property and high abundance of sodium atoms, the sodium layer has been widely observed and studied all over the world (Marsh et al., 2013; Collins et al., 2002; Plane, 2003; Plane et al., 1999)." Here, I need to indicate that there is another reason to the sodium layer has been extensively explored: The sodium atom has large resonant backscatter cross section (especially compare to the cross sections of Fe and Ni atoms). (Collins et al., 2002)
Reply: Thanks for the comment. We have added the cross section around line 7 on page 2 in the revised manuscript.

2. In the last version, the differences of sodium density between Figure 1a and Figure 1b were quite large. Now I understand that the reason is the difference of the two lidar directions. The wide band lidar provides the vertical sodium density, while the narrowband lidar observes the oblique direction. In this version, the authors discard the bad data by the wide band lidar. But many parts in paper still mentioned two lidars. (For example: page1 line 26 and page 3 line23). Please check the manuscript carefully and revise these mistakes.
Reply: Thanks for the comment. We have checked these mistakes throughout the manuscript.

3. The difference between Figure 2 and Figure 5: From Figure 2, the $E_S$ was not present between 13:20 to 14:20. But from Figure 5, the obvious $E_S$ was observed at 14:00. From Fig.5, the $foE_S$ at 1400 UT is about 4.5MHz, and at 1530UT is about 5MHz. But these $foEs$ were not marked in Fig.2.
Reply: Thanks for the comment. We are sorry for the confusing descriptions of these two figures. First, we would like to explain the frequencies recorded by the ionogram. The refractive index (RI) in a cold collisionless plasma is given by the Appleton-Hartree equation:

$$\mu^2 = 1 - \frac{X}{1 - \frac{Y_T^2}{2(1-X)} \pm \sqrt{Y_L^2 + \frac{Y_T^4}{4(1-X)^2}}} \quad , \tag{1*}$$

where $X = \frac{\omega_p^2}{\omega^2}$, with the electron plasma frequency $\omega_p = \sqrt{\frac{Ne^2}{m_e \varepsilon_0}}$ and the incidence wave frequency $\omega$;

$Y = \frac{\omega_B}{\omega}$, with the electron gyro-frequency $\omega_B = \frac{eB}{m}$;

$Y_L = Y \cos\theta$;

and $\quad Y_T = Y \sin\theta$.

When the wave vector $\vec{k} \parallel \vec{B}$ ($\theta = 0$), the signs $\pm$ are referred to the left-hand and right-hand wave with circular polarization.

When $\vec{k} \perp \vec{B}$ ($\theta = \frac{\pi}{2}$), the A-H equation (1)* would be given as:

$$\mu^2 = 1 - X \qquad\qquad (2)^*$$

and

$$\mu^2 = 1 - \frac{X}{1 - \frac{Y^2}{1-X}} \qquad\qquad (3)^*$$

Then the wave determined by equation (2)* is called the ordinary wave (O-wave), which is unrelated to the magnetic field. Equation (3)* corresponds to the extraordinary wave (X-wave), indicating a dependence on the magnetic field because of the value Y. The ionograms of Fig.5 mainly display the ordinary (marked by "O" with pink and red colors) and extraordinary (marked by "X" with dark and light green colors) waves. These wave frequencies are used for calculations of $f_oE_S$, $f_bE_S$, $f_oE$, $f_zE$, $f_xE$, and so on. According to the Handbook of Ionogram Interpretation and Reduction (Section 4, Piggott W.R., and Raver K., 2nd, Elesvier, 1987), foEs equals to the maxima frequency of the O-wave.

Then we checked the original data files carefully, and found the SAO-Explorer gave some wrong values of foEs. We have calibrated the foEs values on 14:00 UT and 15:30 UT again. We subtract all the other modes of waves, and leave the O-wave alone (shown by the following figures). The upper image indicates $f_oE_S$=4.99MHz at 14:00 UT, and the lower picture shows a value of 5.28 MHz at 15:30 UT.

We have modified Fig. 2 in the manuscript, adding the new calculated foEs values.

Thanks again for all the contributions made by this kind and wise reviewer.